# Design of a New Gemini Lipoaminoacid with Immobilized Lipases Based on an Eco-Friendly Biosynthetic Process

Patrícia M. Carvalho [1], Rita C. Guedes [2,3], Maria R. Bronze [3,4], Célia M. C. Faustino [2,3] and Maria H. L. Ribeiro [2,3,*]

1. Instituto de Medicina Molecular, Faculdade de Medicina, Universidade de Lisbon, Av. Prof. Egas Moniz, 1649-028 Lisbon, Portugal; pcarvalho@medicina.ulisboa.pt
2. Research Institute for Medicines (iMed.ULisboa), Faculdade de Farmácia, Universidade Lisboa, Av. Prof. Gama Pinto, 1649-003 Lisbon, Portugal; rguedes@ff.ulisboa.pt (R.C.G.); cfaustino@ff.ulisboa.pt (C.M.C.F.)
3. Department of Pharmaceutical Sciences and Medicines, Faculty of Pharmacy, Universidade Lisboa, Av. Prof. Gama Pinto, 1649-003 Lisbon, Portugal; mbronze@ibet.pt
4. iBET, Instituto de Biologia Experimental e Tecnológica, Avenida da República, Quinta-do-Marquês, Estação Agronómica Nacional, Apartado 12, 2780-901 Oeiras, Portugal
* Correspondence: mhribeiro@ff.ulisboa.pt; Tel.: +351-21-7946453; Fax: +351-21-7946470

**Abstract:** Lipoaminoacids (LAA) are an important group of biosurfactants, formed by a polar hydrophilic part (amino acid) and a hydrophobic tail (lipid). The gemini LAA structures allow the formation of a supramolecular complex with bioactive molecules, like DNA, which provides them with good transfection efficiency. Since lipases are naturally involved in lipid and protein metabolism, they are an alternative to the chemical production of LAA, offering an eco-friendly biosynthetic process option. This work aimed to design the production of novel cystine derived gemini through a bioconversion system using immobilized lipases. Three lipases were used: porcine pancreatic lipase (PPL); lipase from *Thermomyces lanuginosus* (TLL); and lipase from *Rizhomucor miehei* (RML). PPL was immobilized in sol-gel lenses. L-cystine dihydrochloride and dodecylamine were used as substrates for the bioreaction. The production of LAA was evaluated by thin layer chromatography (TLC), and colorimetric reaction with eosin. The identification and quantification was carried out by High Performance Liquid Chromatographer-Mass Spectrometry (HPLC-MS/MS). The optimization of media design included co-solvent (methanol, dimethylsulfoxide), biphasic (*n*-hexane and 2-propanol) or solvent-free media, in order to improve the biocatalytic reaction rates and yields. Moreover, a new medium was tested where dodecylamine was melted and added to the cystine and to the biocatalyst, building a system of mainly undissolved substrates, leading to 5 mg/mL of LAA. Most of the volume turned into foam, which indicated the production of the biosurfactant. For the first time, the gemini derived cystine lipoaminoacid was produced, identified, and quantified in both co-solvent and solvent-free media, with the lipases PPL, RML, and TLL.

**Keywords:** cystine; docking; lipase; lipoaminocids; sol-gel

## 1. Introduction

Biosurfactants show diverse applications in the fields of biomedical, food, cosmetic, agriculture and bioremediation [1], among others, mainly due to their low toxicity, high biodegradability, and multifunctionality. For example, in food, they are used as emulsifiers and preservatives [2]; in cosmetics, they are used due to their lower moisturizing properties and skin compatibility [3]; and in agriculture, they are used to dilute and disperse other compounds like fertilizers [2]. Moreover, among biosurfactants, cationic surfactants are a group that have the capability to disrupt bacterial membranes by combining hydrophobic and electrostatic adsorption [4]. Additionally, due to its detergent-like features, they also compromise the adhesion of pathogenic organisms to the surfaces, proving effective in preventing colonization and infection when used as, for example, a coating in medical insertional materials [3].

Gemini biosurfactants are dimers of the simpler one chained structures with one polar group (Figure 1). This class of amphipathic compounds shows a higher surface activity when compared to single chain structures or conventional surfactants [5], high solubilizing, wetting, foaming capacities, antimicrobial and lime soap dispersion [5]. The supramolecular organization of gemini structures depends on the spacer length, valency, head group size, and tail length. Studies [6] have shown that the spacer length for a good gene vehicle is six methylene groups: two nitrogen atoms prone to be protonated, two unsaturated alkyl tails, and a hydrophilic sugar head.

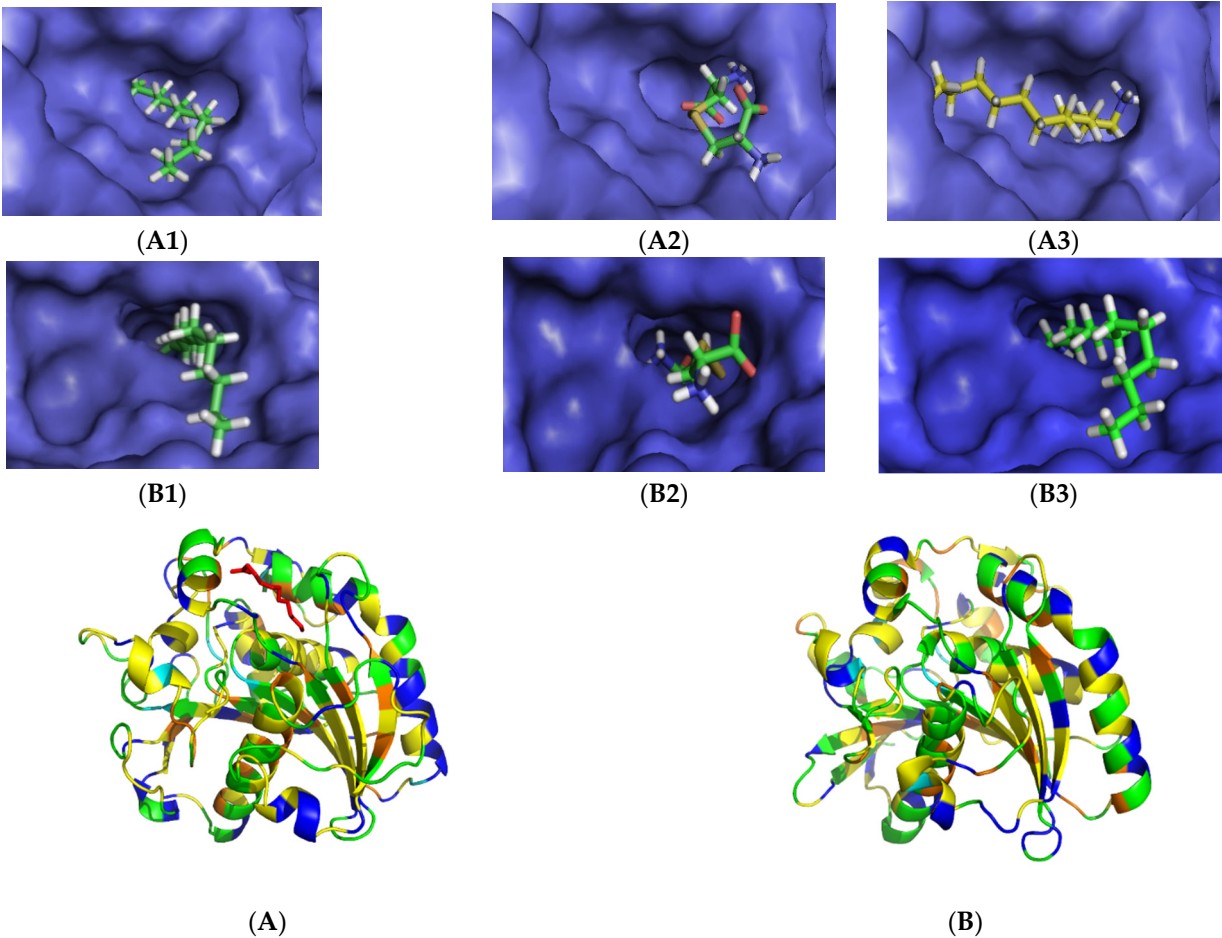

**Figure 1.** *Thermomyces lanuginosus* lipase (TLL) enzyme structure validation through re-docking of lauric acid (**A1**), in which RMSD (Root-Mean-Square Deviation of Atomic Positions) between the predicted and the experimental crystallographic poses less than 2 Å and docking of the same molecule to the active site of the *Rizhomucor miehei* lipase (RML) (**B1**). Docking of cystine (**A2**) and dodecylamine (**A3**) into the active site of TLL. Docking of cystine (**B2**) and dodecylamine (**B3**) into the active site of RML. The position and torsion of dodecylamine is very similar to the one of lauric acid. Crystallographic structures of TLL (**A**) and RML (**B**). The TLL structure is complexed with lauric acid while the RML is uncomplexed.

Lipoaminoacid (LAA) surfactants are gemini biosurfactants formed by a polar head, the amino acid, and a chain of hydrocarbon alkyl as the hydrophobic moiety [5,7]. They can be synthesized from acidic, basic, or neutral amino acids such as glutamate, glycine, alanine, arginine, aspartate, leucine, serine, proline, or even from protein hydrolysates [8]. The amino acid head defines properties like adsorption, aggregation, and biological activity [5]. In the medical sector, many lipoamino acid/peptides are known to have antibiotic, antiviral [9], or even antitumor activity, and to have the ability to modulate the immune system or inhibit some enzymes and toxins [3]. Additionally, the combination of LAA activity with nucleic acids and the ability to form complex aggregates are promising candidates in the field of gene therapies as a new vehicle for delivery inside the cell [4]. Spontaneous

formation of complexes between cationic lipids and DNA are called lipoplexes and have special interaction features with the plasma membrane due to electrostatic interactions, facilitating adsorption [10,11]. The arrangement of lipoplexes is identified by combining small-angle x-ray scattering and electron microscopy [11,12]. Studies have revealed that gemini structure surfactants show a greater efficiency compacting DNA [2,6] than other biosurfactants.

The critical micellar concentration (CMC) is a parameter related to hydrophobicity [5] that represents the concentration above which monomeric surfactant molecules assemble into aggregates, called micelles [13]. The CMC levels for gemini structures are up to two orders of magnitude lower than the respective linear surfactant [5], which therefore increases their efficiency in reducing the surface tension of water [9].

Biosurfactants can be produced through a biosynthetic process, using more competitive substrates that are eco-friendly, biodegradable [14], and show low toxicity, among others properties [9]. In fact, a biocatalytic process offers several advantages such as mild reaction conditions, selectivity, direct performance of a reaction on a substrate, reduced or inexistent toxic sub-products, which consequently has better results in product separation and purification [7]. Lipases are biocatalysts that do not need cofactors, are not too costly, thus they contribute to make the overall process less expensive than conventional chemistry and all the costs associated with time to industry.

In the context of this work, a gemini LAA was developed by the enzymatic route with lipases using a cystine based approach.

Lipases are naturally involved in lipid and protein metabolism and are currently one of the most widely used enzymes in biotechnology [15–17]. Lipases (EC 3.1.1.3, triacylglycerol lipase) are water-soluble hydrolases, which can be found in animals, plants, and microorganisms [18]. In the catalytic domain of these enzymes can be found the active site and lipid binding determinants. At the core of this catalytic domain, there is an $\alpha/\beta$-hydrolase fold, which basically translates into a motif of $\beta$-sheets connected by $\alpha$-helices, chemically similar but structurally different to what is found in serine proteases [18] and supporting a catalytic triad of Ser-His-Glu/Asp [19]. The serine is inserted in a motif of stand-turn-helix in stand $\beta5$ with unusual $\Phi$ and $\Psi$ torsion angles. This forms a sharp turn, highly conserved due to its functional relevance, called the nucleophile elbow [20].

Typical substrates are mainly water-insoluble apolar substrates like fats and oils, which have long hydrocarbon-like groups that are mostly flexible, and a wide variety of these molecules are accommodated in the active site of the enzyme. Specificity usually differs in preference of acyl and alkyl group size, but some, like *Rizhomucor miehei* lipase (RML), have a strong stereospecificity while others, like porcine pancreatic lipase (PPL), do not [19]. This specificity is categorized by the position of ester bonds (hydrolyzed or formed) in the substrate molecules (regioselectivity), the class of substrate accepted (chemo-selectivity), and the stereoselectivity [20].

Organic solvents have been an option to solubilize apolar substrates, which follow some prerequisites, like (i) not being a substrate for lipase, (ii) not interfering in the reaction by inactivating the enzyme [21], and (iii) allow easy separation from the final product. Some studies suggest that solvents with a log P lower than 3 are preferable to the bioreaction by lipases [21]. As eutectic mixtures can be a "green" alternative to organic solvents, they can be tested in this biocatalytic system where they combine a solid organic salt, like sodium chloride, and a complexing agent, like urea or glycol [22]. Eutectic liquids tend to cost about the same as organic solvents and because purification processes for salt removal can be discarded, the process can become even cheaper [23]. These solvents have been shown to be appropriate for enzymatic reactions with lipases, maintaining enzyme stability and increasing activity [22]. A high stability of the enzyme in the solvent used could allow a rise in temperature, thus, increasing the catalysis rate [23].

The main disadvantages in the use of biocatalytic systems are the cost of some enzymes and their mixing with the final products, slow reaction rates, and the absence of optimized reactors for biphasic media catalysis [24]. Nonetheless, the immobilization of lipases can

overcome some of the drawbacks. The choice of the solid matrix is crucial to provide a good operational and thermo stability [21]. Another requisite for immobilization is that the method should be cost-effective, reproducible, and should pay attention to the grade of the materials and equipment used [25].

The main forms of commercial immobilized lipases are by adsorption onto polymer based matrices, nonetheless Novozymes has also developed a new and less expensive method by combining adsorption and encapsulation in silica granules. Additionally, when the immobilized particles are also magnetic, the enzyme becomes easy to recover and the overall process becomes even more productive.

Another critical point concerning the reaction, specifically involving lipase, is the conformational changes. Lipase has two different conformations, closed (inactive) and open (active), so it is very important to ensure that the enzyme is immobilized in its active form to guarantee good reactions rates [26,27]. The strategies to immobilize the enzyme in an active conformation are the use of hydrophobic materials or the use of detergents during the method [28]. The use of hydrophobic matrices increases the lipase activity in most of the immobilization protocols.

The main goal of this work was the development of a new cystine derived gemini lipoaminoacid biosurfactant using an eco-friendly production system based on immobilized lipases, toward future specific applications in gene and drug delivery. To attain the main goal, specific intermediate milestones were designed, namely (i) docking studies; (ii) screening the best lipase for the biosynthesis; (iii) development of an efficient method of lipase immobilization for the biocatalytic process; (iv) effect of reaction conditions (e.g., temperature, reaction time-course and operational stability) in lipoaminoacid surfactant production; and (v) design and optimization of the bioreaction media.

## 2. Results and Discussion

### 2.1. Substrate Docking Studies in Rizhomucor miehei Lipase (RML) and Thermomyces lanuginosus Lipase (TLL) Active Sites

In order to evaluate the fitting of the substrates cystine (Cys) and dodecylamine (Dda) to the active site of lipases (RML and TLL), first, site docking studies were carried out to produce the gemini lipoaminoacid biosurfactant. This methodology predicts the possible interactions between the ligand and receptor, generating several docking solutions by altering the position and torsion of the ligand [29].

In these studies, a crystallographic structure of TLL complexed with lauric acid was used as a model, since it is also a small linear molecule similar to the substrates. The crystal structure-based model, after the removal of waters and other non-residue molecules from the coordinate set, was validated by re-docking of the lauric acid. The experimental pose of lauric acid was reproduced [Figure 1(A1)]. The RMSD (Root-Mean-Square Deviation of Atomic Positions) between the predicted and experimental crystallographic poses was less than 2 Å, which was quite satisfactory considering the rotatable bonds of the lauric acid molecule.

RML was considered uncomplexed. The docking in the RML model and the lauric acid molecules showed a different fitting [Figure 1(B1)], not entering as deep into the active pocket and presenting a changed rotation.

After enzyme structure validation, the dodecylamine and cystine molecules were docked, with both substrates fitting in the active site pocket of both enzymes (Figure 1A,B).

The TLL pocket maintained its structure while accommodating the substrates, cystine and dodecylamine, taking advantage of deeper clefts of the active site [Figure 1(A2,A3)]. Comparing the position of the dodecylamine with the reference pose of the lauric acid, it was observed that the first preferably adapted to the opposite cavity of the catalytic pocket, which were also the cystine niches.

Comparing the TLL and RML docking results, some differences was observed [Figure 1(A2,A3,B2,B3)].

The docking of cystine in RML was done on the opposite side of the deeper cleft as well as in a different rotation (Figure 1(B2)). The docking of Dda in the RML active site

[Figure 1(B3)] produced a position and torsion of the molecule very similar to the one of lauric acid, and it was very interesting to see the molecules behaving in the same way.

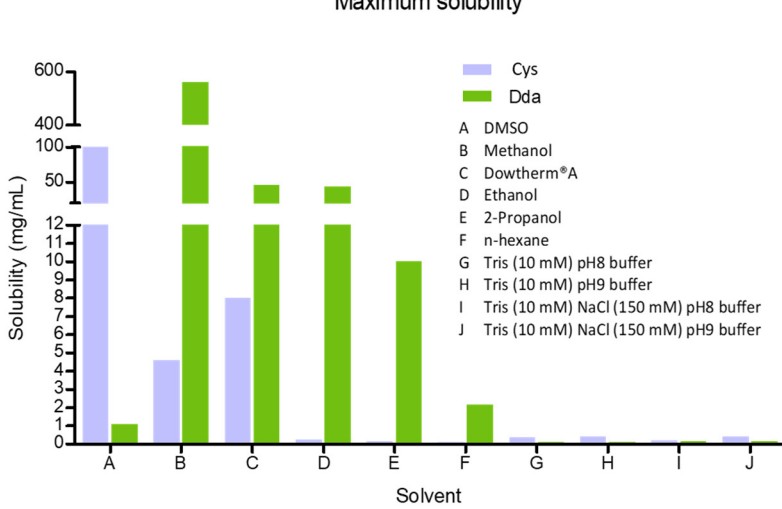

**Figure 2.** Solubility of the substrates (mg mL$^{-1}$), cystine (Cys) (blue) and dodecylamine (Dda) (green), at 40 °C.

Nonetheless, these results raise challenges relating to the exact mechanism of the reaction. As some authors [30,31] have stated, the understanding of the intermediates and their positioning in the active site is often a key point to the successful prediction of enzymatic reactions.

Lipases are enzymes, where the active site structure is a conserved structure. In this study, the substrates (Cys and Dda) were docked in a TLL model. However, both RML and TLL are fungal lipases that authors usually group together when doing structural analysis. For example, other authors [32] have grouped TLL and RML in the same group of active site conformation when studying small acid/alcohol ligands because both allow a large alcohol part substrate, but more limited/narrow acyl binding cleft on the acid part.

The docking prediction method for lipases is challenging to apply. These structural studies are easily applied to enzymes with highly charged moieties present in the active site and actively participate in the modeling of the substrate to a productive conformation and in the creation of high docking energy differences. Metalloenzymes are a good example of this because the metal cations are highly charged and have strong interactions with the ligand [29], but lipases do not possess any metal cations near the active site. All lipases show a common hydrolase fold and a catalytic triad composed of a nucleophilic serine, which is activated via hydrogen bonds as part of a charge relay system, along with the histidine and the aspartate or glutamate residues. Additionally, TLL has four tryptophan residues located in positions 89, 117, 221, and 260. Due to their relatively large size and two aromatic rings, these last residues can take part in most of the special conditioning and be a determinant in the interactions between the ligands and the enzyme.

Another challenge in these simulation studies are the available reference structures, which are typically determined by X-ray or Nuclear Magnetic Resonance (NMR) spectroscopy in an aqueous environment. In the particular case of lipases, there are conformational changes between aqueous and organic media. Nonetheless, some studies [29] with small molecule ligands have found a small difference in the simulation results when predicting for media alterations.

In conclusion, the docking studies demonstrated the fitting of the substrates cystine and dodecylamine to the active site of the lipases RML and TLL.

As can be seen (Figure 1), there was docking of cystine (A2) and dodecylamine (A3) into the active site of TLL and docking of cystine (B2) and dodecylamine (B3) into the active

site of RML. The position and torsion of dodecylamine was very similar to the one of lauric acid.

Crystallographic structures of TLL (A) and RML (B). The TLL structure was complexed with lauric acid while the RML was uncomplexed. Hydrophobic regions showed as yellow/orange patches ("dry desert" colors, hydrophobic) on the protein surface, and hydrophilic regions as green and blue ("wet, watery" colors, hydrophilic).

## 2.2. Production of a Gemini Lipoaminoacid Biosurfactant

In the production of LAA, the main problem in using enzymes as lipases is that the product is the result of a bioconversion of two substrates with different polarities. In fact, the amino acid is hydrophilic and the tail is hydrophobic, therefore the reaction (Scheme 1) depends on the choice of a suitable solvent for the two components, which was studied in the next part for cystine and dodecylamine.

L-cystine                                         R = (CH$_2$)$_{11}$CH$_3$

**Scheme 1.** Biosynthesis of gemini lipoaminoacid base on enzymatic reaction with lipases/proteases using the substarte L-cystine and dodecylamine.

Of the three lipases tested in this work, two were commercial immobilized in a granular form in macroporous ion-exchange resin and in silica, respectively, TLL and RML, and one was a free form, porcine pancreatic lipase (PPL). Afterward, this lipase was immobilized in-house in advanced matrices, sol-gel PPL lenses, with different compositions and shapes of the particles.

### 2.2.1. Solvent Choice for Substrates

Cystine and dodecylamine have opposite polarities and very low solubility in water. The choice of the best solvents for both substrates for lipases is an important achievement in the bioconversion media design. Therefore, substrate solubility assays were assessed in (i) aqueous systems, (ii) organic solvents, and (iii) an eutectic mixture. In the first case, two buffers, Tris and Tris plus NaCl, at pH 8 and pH 9, were used, based on the studies of [33], which stated that at higher pH and osmolarity, the solubility of cystine increased. The organic solvents tested were ethanol, 2-propanol, *n*-hexane, dimethyl sulfoxide (DMSO), methanol.

Cystine

The polar heads of gemini LAA will consist of cystine, which presents a low solubility in water but also in organic solvents. In an aqueous system, the solubility of cystine was very similar at pH 8 and pH 9, respectively, 0.38 and 0.40 mg mL$^{-1}$ (Figure 2). The addition of NaCl to the buffer, pH 8, led to a cystine solubility of 0.19 mg mL$^{-1}$ (Figure 2) and the buffer, pH 9, to a solubility of 0.42 mg mL$^{-1}$ (Figure 2). This last result is in line with the published studies of [33], where the solubility of L-cystine increased with the amount of salt in the aqueous solution and with the increase in pH.

Ethanol, 2-propanol, and *n*-hexane were not good as solvents for cystine, with solubility concentrations of about 0.22 mg mL$^{-1}$. Among the other solvents tested for cystine, the best was DMSO, with a solubility around 99.68 mg mL$^{-1}$, followed by methanol, however, it was twenty times lower, at 4.58 mg mL$^{-1}$ (Figure 2).

Dodecylamine

Dodecylamine is a compound with a highly lipophilic carbonated chain compound with low solubility in water. Among all the solvents tested, the aqueous buffers clearly had the lowest solubility. In Tris buffers, pH 8 and pH 9, dodecylamine was almost insoluble, however, the addition of NaCl to both buffers increased the solubility to 0.14 mg mL$^{-1}$ (Figure 2).

The minimum solubility of dodecylamine was in DMSO at 1.07 mg mL$^{-1}$, being twice in $n$-hexane, 2.16 mg mL$^{-1}$ (Figure 2). The best results were obtained in 2-propanol, 10 mg mL$^{-1}$, in ethanol, 43 mg mL$^{-1}$, and in methanol, 560 mg mL$^{-1}$, the highest solubility (Figure 2). Organic solvents were the best to solubilize dodecylamine.

Comparing the solubility of each substrate in the solvents tested (Figure 2), it was possible to conclude that the organic solvents worked well for dodecylamine, while cystine had a lower solubility. In contrast, in aqueous solutions, the solubility of cystine was better than for dodecylamine. These values clearly reflect the different polarities of the substrates. Only in DMSO was cystine more soluble than dodecylamine.

Based on the results shown in Figure 2, it is possible to conclude that individually, the best solvents were DMSO for cystine (99.68 mg mL$^{-1}$) and methanol for dodecylamine (560 mg mL$^{-1}$), however, the solvent that compromised a better solubility for both substrates was the eutectic mixture Dowtherm$^{\circledR}$A in which cystine dissolved at 8 mg mL$^{-1}$ and dodecylamine dissolved at 46 mg mL$^{-1}$.

In a reactional scenario, the substrates used a 1:2 (cystine:dodecylamine) proportion, so in the further solubility assays, this ratio was used in the production of gemini LAA.

2.2.2. Development and Characterization of Sol-Gel Lenses for Porcine Pancreatic Lipase (PPL) Immobilization

The several methods tested for the preparation of PPL sol-gel lenses are described in Section 3.6, following the reactions defined in Table 1. Briefly, two different methods for hydrogel formation, with different buffers (Tris and acetate), and two pH values (6 and 9) (Table 2). The goal was to evaluate the effect of different salts and/or different pH values in the matrix formation. Acetate buffer pH 6 was chosen based on their application for the immobilization of other enzymes [34]. Tris buffer was tested at pH 9.

**Table 1.** Reactions that describe the sol-gel events.

$$\equiv\text{Si–OR} + \text{H}_2\text{O} \underset{\substack{\longleftarrow \\ esterification}}{\overset{\substack{hydrolisys \\ \longrightarrow}}{}} \equiv\text{Si–OH} + \text{ROH}$$

$$\equiv\text{Si–OR} + \text{HO–Si}\equiv \underset{\substack{\longleftarrow \\ alcoholysis}}{\overset{\substack{alcohol\ condensation \\ \longrightarrow}}{}} \equiv\text{Si–O–Si}\equiv + \text{ROH}$$

$$\equiv\text{Si–OH} + \text{HO–Si}\equiv \underset{\substack{\longleftarrow \\ hydrolysis}}{\overset{\substack{water\ condensation \\ \longrightarrow}}{}} \equiv\text{Si–O–Si}\equiv + \text{H}_2\text{O}$$

The different methodology tested allowed for the production of lenses with very good yields (Figure 3), except for the A1 methodology. In fact, A1 lenses were made using Tris buffer at pH 9, while in all the others, it was used as a buffer at pH 6. The lenses in A1 started cracking early during the drying stage and most of them remained stuck to the microplate well (Figure 3). However, when pH 6 buffers were used, the lenses remained intact with minimal differences between the use of Tris or acetate buffers, and a simple inversion of the microplate was sufficient to collect most of them. This result shows the importance of the pH in the formation of the matrix.

**Table 2.** Preparation of PPL lenses according to (i) method A, in which the hydrogel parts are only joined in the microplate well; (ii) method B, in which the hydrogel parts are first joined in the microtube and then transferred to the microplate well.

| | | | |
|---|---|---|---|
| **Method A** | | | |
| A1 | 25 µL Sol | + | 25 µL PPL 2.5 mg mL$^{-1}$ in 10 mM Tris buffer pH 9 |
| A2 | 25 µL Sol | + | 25 µL PPL 2.5 mg mL$^{-1}$ in 10 mM Tris buffer pH 6 |
| A3 | 25 µL Sol | + | 25 µL PPL 2.5 mg mL$^{-1}$ in 20 mM acetate buffer pH 6 |
| **Method B** | | | |
| B1 | Sol | + | 500 µL PPL 2.5 mg mL$^{-1}$ in 10 mM Tris buffer pH 6 |
| B2 | Sol | + | 500 µL PPL 2.5 mg mL$^{-1}$ in 20 mM acetate buffer pH 6 |

Measurements of the diameter and thickness were taken for the yield of lenses produced by each formulation (Figure 3).

The aging of the matrix, when drying in the microplate well, greatly influenced the access to the enzyme. The more spread-out the matrix, the bigger the diameter and consequently less thick, which means more available surface and less matrix resistance with the transference of substrates and product in the support facilitated. The data shown in Figure 3, related to the different methods of hydrogel formation, method A and B, respectively, produced in each well individually and as a whole in the microtube, did not greatly influence the lenses, compared to the type of buffer used. The value for each parameter was always similar between A2-B1 (Tris buffer), and A3-B2 (acetate buffer) (Figure 3). The substitution of the acetate buffer for Tris buffer (A2 and B1) originated lenses were less compact, more spread-out, and thinner, which could facilitate the transport of substrates and product in and out of the matrix. In Figure 3, it can be seen that by using this protocol, the lenses produced were smaller in diameter but thicker than the ones where Tris buffer was used. In conclusion, for the immobilization of PPL lipase in sol-gel lenses, Tris buffer exhibited better results.

Among all the parameters studied, the weight was the one that experienced reduced modifications with hydrogel protocol changes or between Tris and acetate buffers, but lenses B1 still had the lowest standard deviation value, assuring a greater precision in the amount of enzyme contained in each lens. In conclusion, B1 lenses, produced with Tris buffer at pH 6, allowed us to obtain high yields and reduced mass transfer conditions due to the exposure of a greater surface.

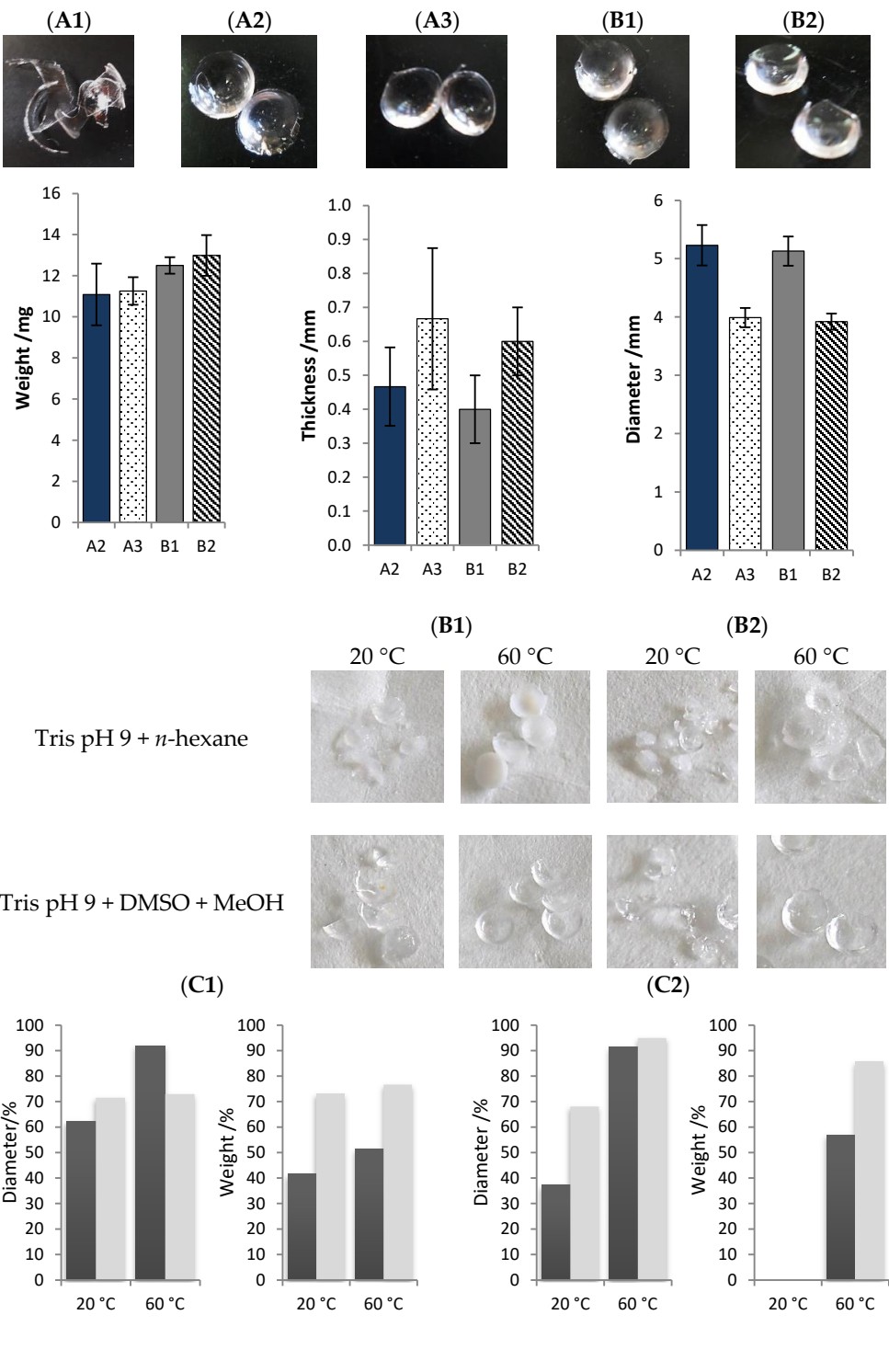

**Figure 3.** Diameter, weight, and thickness parameters of the lens produced according to methods **A** or **B**, with the enzyme in Tris buffer (**A2** and **B1**) or acetate buffer (**A3** and **B2**). The A1 lens fractured and measurement was not possible. Sol-gel matrices' resistance to the organic solvent combinations used as reaction media was assessed by the percentages of diameter and weight, after incubation in buffer Tris pH 9:*n*-hexane (**C1**) and DMSO:MeOH (**C2**) for 24 h at 20 and 60 °C. The lenses used in these assays were produced by the methods B1 and B2.

2.2.3. Evaluation of the Behavior of the Sol-Gel Immobilization Matrix to Solvents

As the bioproduction media consisted of biphasic and co-solvent systems, it was important to evaluate the compatibility of the sol-gel matrix, where PPL was immobilized, with the different media.

Therefore a biphasic system of Tris buffer and *n*-hexane was tested and a co-solvent system of DMSO and methanol, the two best solvents for cystine and dodecylamine individually, respectively. Only the hydrogel preparations obtained by method B were used in these assays.

In the presence of *n*-hexane, the lenses turned opaque, almost with a white coloration (Figure 3), and became very fragile, while in DMSO, the lenses tended to maintain a translucent aspect.

Figure 3(C1,C2) presents the percentage of each parameter analyzed just before collecting the particles from a 24 h batch in each mixture. The diameter measurements led to the conclusion that C1 lenses had a more resilient matrix. The older batch, incubated at 20 °C, held between 60 and 70% of the original diameter in both solvent mixtures, while for C2 lenses, this value was lower than 40% in the presence of *n*-hexane. Moreover, it was possible to conclude that due to the observed changes in the aspect of the matrix, the biphasic media was too aggressive for this kind of immobilization and a DMSO/MeOH system was preferable to the maintenance of the matrix integrity.

2.2.4. Bioreaction Media Design

In order to develop and optimize biocatalytic systems for the production of new lipoaminoacids using immobilized lipases with a high activity, several experiments using media with different characteristics and enzyme concentrations were studied.

First, the liquid state reactions were mainly monitored by thin layer chromatography (TLC) assays in the conditions previously optimized. The analytical assays occurred using chloroform:methanol (7:3) eluents, stained with ninhydrin and Dragendorff reagents. TLCs analyses were carried out from the reaction samples at 0, 15, 30, 180, and 300 min.

Quantification of the bioproduct in the several bioreaction media was carried out by HPLC-MS/MS. Multiple reaction monitoring (MRM) assays were applied to the samples in order to identify and quantify the new gemini lipoaminoacid in the different media.

Aqueous Media

PPL was tested at several concentrations (0.35, 0.87, 1.75, and 7 mg mL$^{-1}$) in aqueous media of Tris buffer at pH 9. In the TLC analysis for the different reactions, in the ninhydrin stains was noticed a new spot in comparison to the control substrates. The biocatalysts, lipases RML and TLL, were tested in a concentration of 3.3 mg mL$^{-1}$. As observed in the PPL reaction, a stained spot was also present in the TLL and RML assays, which indicated the presence of a new compound, a lipoaminoacid.

The time-course and the thermostability of lipase RML was evaluated in the aqueous Tris buffers at pH 8 and pH 9 in three consecutive batches of 24 h each. This study allowed for the establishment of a working temperature where the lipase was active and the matrices resistant. For all three lipase concentrations tested, at pH 8 and 40 °C (Figure 4), the absorbance was higher than at 50 and 60 °C. Moreover, the best results were attained at pH 9.

(**a**)

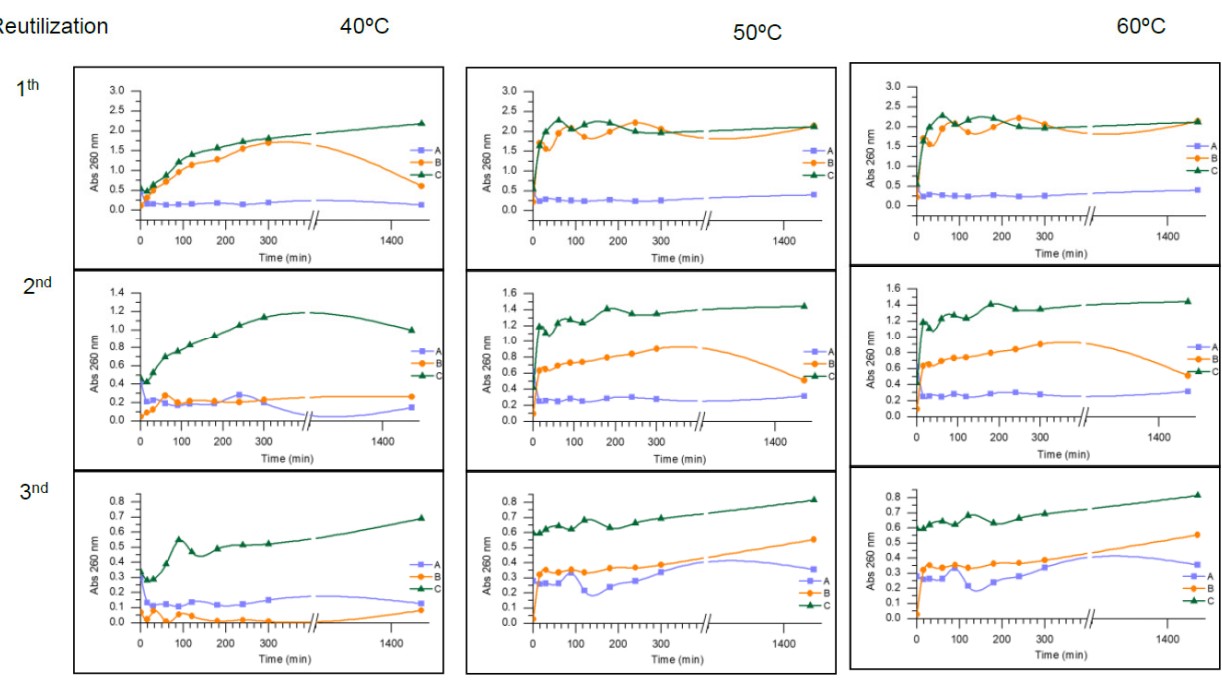

[Enzyme] A- 0.5 mg mL⁻¹; B − 2 mg mL⁻¹; C − 4 mg mL⁻¹

(**b**)

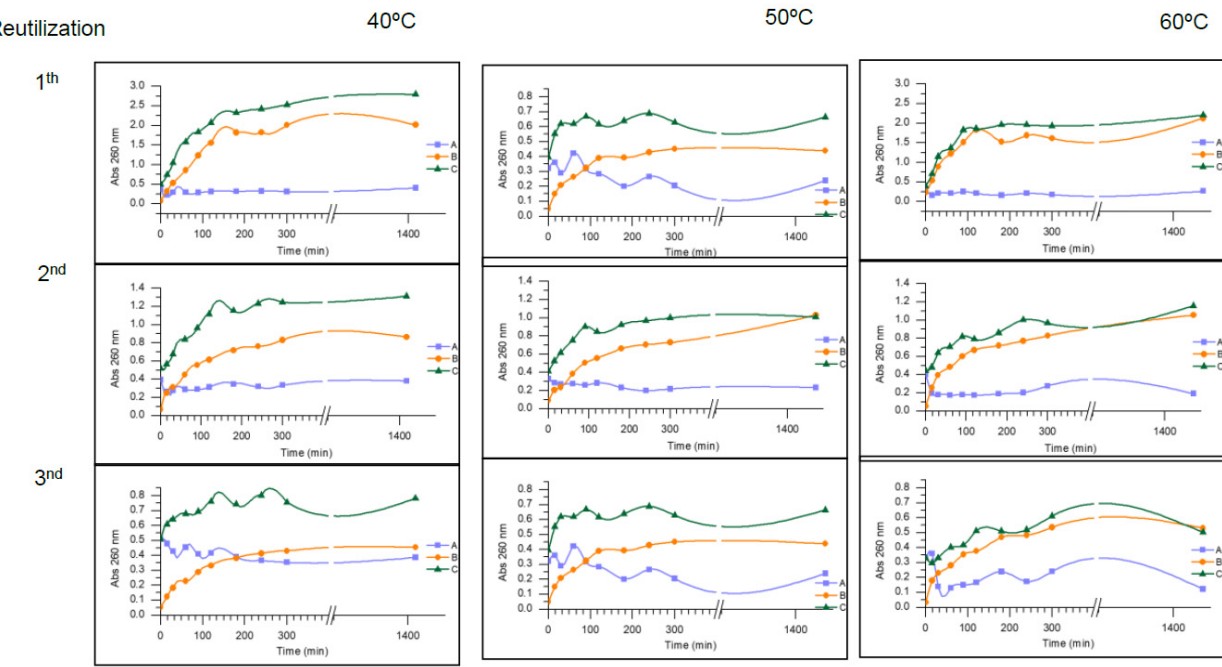

[Enzyme] A- 0.5 mg mL⁻¹; B − 2 mg mL⁻¹; C − 4 mg mL⁻¹

**Figure 4.** Time-course and thermostability of Lipozyme® RML, in Tris buffer 10 mM, pH 8 (**a**), pH 9 (**b**). Lipase RML concentration was 0.5 mg mL⁻¹ (A), 2 mg mL⁻¹ (B), 4 mg mL⁻¹ (C). The SD (Standard Deviation) was ±0.01. Each point of the graphic was carried out in triplicate.

The residual activity of lipase RML at pH 8 after three consecutive reutilizations of 24 h each batch was around 35%, at the temperatures 40, 50, and 60 °C, while at pH 9, almost 75% residual activity remained at 50 °C (Figure 5).

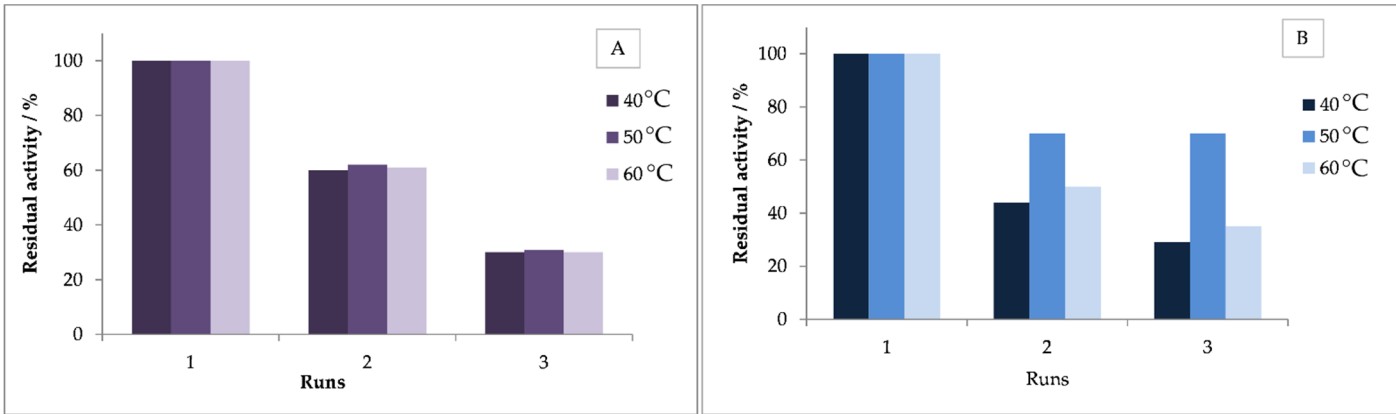

**Figure 5.** Residual activity of Lipozyme® RML in aqueous system. (**A**) Tris buffer 10 mM, pH 8; (**B**)Tris buffer 10 mM, pH 9.

Co-Solvent Systems

Different co-solvent reaction systems were tested in the production of a new lipoaminoacid using the biocatalysts TLL, RML, and PPL, with the substrates cystine and dodecylamine.

The three lipases were tested in the media of Tris buffer pH 9:2-propanol (2:1). The analysis of the samples from these reactions show the appearance of a new spot on the top of the TLC plate, which meant catalytic modifications in all situations, with the formation of a new compound. The reactions were also monitored directly at 260 nm and by the Bradford method. There was no detection of protein in the Bradford assays for the reactions in this medium. RML showed an evolution pattern of the reaction in time, similar to the ones obtained in aqueous media.

In the co-solvent system of DMSO:MeOH using PPL in free and immobilized forms, there was no evidence of any reaction that would lead to the formation of a lipoaminoacid.

When the same media was tested with RML and TLL, a reduction on the amount of substrates was observed. Additionally, in the TLL reaction, the characteristic dragging produced by the dodecylamine disappeared and a new form of light pink was developed.

For the TLC sample analysis in the DMSO:MeOH media, the plate showed a marked area for the reaction with TLL consistent with the production of the gemini lipoaminoacid, with the consequent consumption of the substrates. The results showed that TLL and RML in DMSO:MeOH were successfully used in catalyzing a new gemini lipoaminoacid production.

The patterns produced in the co-solvent system of Tris buffer:DMSO:MeOH (12:1:1) were very similar to the ones verified for Tris buffer:2-propanol.

The identification and quantification of the new gemini lipoaminoacid was carried out afterward by HPLC-MS/MS. In the co-solvent media, the DMSO:MeOH media allowed for the best concentration (7.1 mg/mL) of the new gemini lipoaminoacid produced using the lipase TLL (Figure 6). These results confirm the previous results obtained by TLC analysis. However, in the same co-solvent media (DMSO:MeOH) with the RM lipase, no product was detected. One explanation for this result may be due to the type of immobilization of RML, which did not confer the enzyme with an appropriate conformation to perform the biosynthesis. This result indicates that the enzyme structure and its interaction with the surrounding media are key points in this biocatalysis.

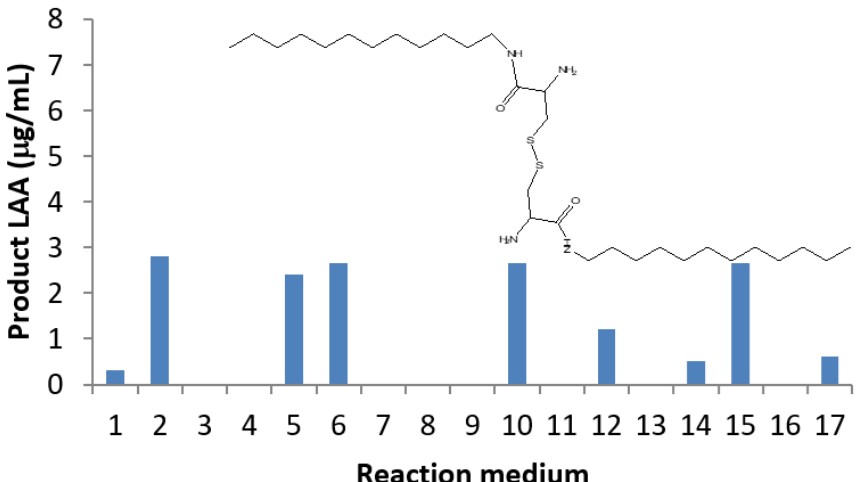

**Figure 6.** Concentration of the new gemini lipoaminoacid (LAA) produced in each bioreaction media design, respectively: 1—RML in Tris buffer pH 9 + *n*-hexane (organic phase); 2—RML in Tris buffer pH 9 + *n*-hexane (aqueous phase); 3—RML in DMSO:MeOH; 4—TLL in Tris buffer pH 9 + *n*-hexane (organic phase); 5—TLL in Tris buffer pH 9 + *n*-hexane (aqueous phase); 6—TLL in DMSO + MeOH; 7—PPL in DMSO:MeOH; 8—Control RML in eutectic mixture without substrate; 9—TLL in eutectic mixture; 10—PPL in eutectic mixture; 11—Control (without enzyme) in eutectic mixture; 12—RML in eutectic mixture; 13—TLL in eutectic mixture; 14—PPL immobilized in sol gel used in eutectic mixture; 15—RML in eutectic mixture (evaporated, solid); 16—TLL in eutectic mixture (evaporated, solid); 17—PPL immobilized in sol-gel used in the eutectic mixture (evaporated, solid).

Biphasic Systems

The biphasic system of Tris buffer pH 9:*n*-hexane (2:1) was tested using different concentrations of Cys and Dda. The samples were applied in the TLC plates separating the aqueous from the organic phases. Comparing the control plates with the PPL reaction plates, a consumption of dodecylamine and the appearance of a specie with a bigger migration rate was noticed. The aqueous phases of these last reactions (2.71/0.27 mg mL$^{-1}$) were monitored by spectrometry at 260 nm and the Bradford method, aside from TLC.

From the TLC for the different samples from RML and PPL, there were significant differences compared to the control, with the samples from RML showing the absence of the dodecylamine area. Using higher reaction volumes, the changes in the interphases were visible for both biocatalysis with TLL and RML enzymes. There was a high intensity of an emulsion concentrating particularly in the interphase of the TLL reaction. These interphase emulsions indicate the formation of a compound with surfactant characteristics. When the lipase (TLL and RML) concentrations increased two- and five-fold, similar emulsions were observed.

Regarding the PPL in both free and immobilized forms, no significant changes were observed. The product from free PPL at 0.31 mg mL$^{-1}$ was not substantially detected with the TLC method.

In the biphasic media of Tris buffer:*n*-hexane, the bioproduction of the gemini lipoaminoacid was accomplished by RML and TLL at concentrations of 2.8 and 2.4 mg/mL, respectively. This result may be due to the immobilization of the enzymes, which stabilizes the enzymes in an active form, and the presence of an interphase further activates the lipases. In this study, it was possible to conclude that in the biphasic system of Tris buffer pH 9 and *n*-hexane (2:1), RML and TLL catalyzed the biosynthesis of the new gemini lipoaminoacid, which was identified and quantified in the aqueous phase.

In conclusion, the increase in TLL and RML concentrations in a biphasic system, from the initial 2.71 to 27 mg mL$^{-1}$, did not initiate changes in the biocatalytic activity.

Solvent-Free Media

Solid-to-solid reactions are particularly interesting because the technology is evolving in the direction of using less solvents and reducing waste. In this type of reaction, the substrates are in suspension and as the product is biosynthesized, it precipitates. These methods are known to create very high yields and sometimes facilitate product recovery [35,36]. The study by [37] showed success in performing biocatalysis in RML and TLL melts at extreme temperatures, where the hydration layer surrounding the enzyme, which even persists in most organic solvent media, was eliminated, forcing the enzyme to react in the direction of esterification and transesterification.

In this work, a similar form was performed on reducing the solvent ratio in the biocatalysis, not by creating enzymatic melts, but by melting one of the substrates. The substrate, dodecylamine, had a melting point of 27–29 °C (according to the manufacturer). Since substrate solubility is an issue in the reaction, it is in the absence of solvents. At the temperature (40 °C) in which these reactions were performed, dodecylamine was liquid, so cystine and lipase were just added to it. Dowtherm® A (55%) helped to unify the whole mixture. After 5 h, all immobilization matrices were resilient enough to resist this type of media.

Both reactions with either the biocatalysts RML or TLL showed equal consistency to the one displayed by the control tube. In this tube, the texture resembled that of "melted soap" or wax, forming a very white and pasty mixture as well as in the reactions with RML and TLL. In the reaction where PPL lenses were used, most of the reactional volume turned into a foam. The modification of the texture consistency verified in this reaction using PPL meant that biosurfactant bioproduction was accomplished at high reactional rates. The product was obtained purified in a solid form. The TLC plates at 5 h of reaction showed that the PPL reaction resulted in high modifications when compared to all the others, and no substrates were detected while a high spot appeared, which corresponded to a new gemini cystine-derived lipoaminoacid biosynthesized. The differences were even more striking when the comparison was made using the staining with the Dragendorff reagent. In this case, all the plates displayed two spots, except for the PPL reaction. In this last plate, the staining agent reacted only with the smaller area, which had the higher migration rate. There were no spots attributed to the substrates. Therefore, according to the TLC analysis, all of the substrates were converted to product.

These samples were also analyzed by the eosin method. When the eosin solution was in contact with the foam from the PPL reaction, it resulted the formation of agglomerates.

For all the other enzymatic reactions, the eosin tubes remained an orange color while in the PPL reaction, the solution changed to a pinker tone. A maximum peak was detected at 538 nm and at lower concentrations of gemini LAA, a correlation with absorbance was established, leading to a calibration curve.

The addition of more eosin allowed the precipitation of the gemini LAA visible to the naked eye. At this concentration, the surfactant agglomerated around the dye particles, clearing the rest of the solution. The entire color was concentrated in these removable particles, supporting the works where these types of surfactants are being studied to clear polluted water streams [38].

Others authors [39–41] have also studied the interaction of eosin with cationic surfactants using fluorescence spectra detecting peak shifts directly related to the surfactant critical micellar concentration (CMC).

In the "solvent free" reaction, the substrate, cystine, was undissolved and the contact with the enzyme could enable its activation. For this system, both RML and PPL sol-gel lenses were able to perform the biosynthesis of the new gemini lipoaminoacid in concentrations of 1.2 and 4.2 mg/mL, respectively. Moreover, while the other samples were prepared by evaporation of the total reactional volumes, the solid system samples were developed by preparing 1 mg mL$^{-1}$ solution of the reaction mixture. Moreover, as the solid reaction media weighed approximately 1300 mg (1 mg corresponds to 0.08% of the reaction medium), in order to compare the reaction product to the other results, it was

assumed that the real value of the gemini LAA in that media, with PPL sol-gel lenses, was 5 mg/mL.

In the solvent free media in which PPL was employed, the reaction displayed modifications consistent with the formation of a new surfactant species, visible by texture changes of the reactional volume to the form of foam. In conclusion, this media design was the best to produce LAA among the different bioreaction media studied.

The results presented in this work highlight the potential of the biocatalytic production of lipoaminoacids, a friendly process, as a suitable alternative to current chemical process of multiple extractions, leading to LAA in high purity.

For the first time, using an enzymatic approach, cystine derived gemini cationic biosurfactants were biosynthesized.

## 3. Experimental

### 3.1. Enzymes

Three lipases were used: porcine pancreatic lipase, Type II, Crude (PPL) (Sigma-Aldrich L3126, E.C. 3.1.1.3), Lipozyme® TL IM (TLL) (from *Thermomyces lanuginosus*) (Novozymes), and Lipozyme® RM (RML) (from *Rizhomucor miehei*) (E.C. 3.1.1.1, Sigma-Aldrich, Saint Louis, MO, USA). TLL are RML are commercially immobilized forms in a macroporous ion-exchange resin, and in silica, respectively.

### 3.2. Substrates

L-cystine (Cys2) (Sigma-Aldrich, Saint Louis, MO, USA) (Figure 7) and dodecylamine (Dda) (Sigma-Aldrich) (Figure 8) were used as substrates to create the polar heads and the hydrophilic tails of the gemini surfactants, respectively.

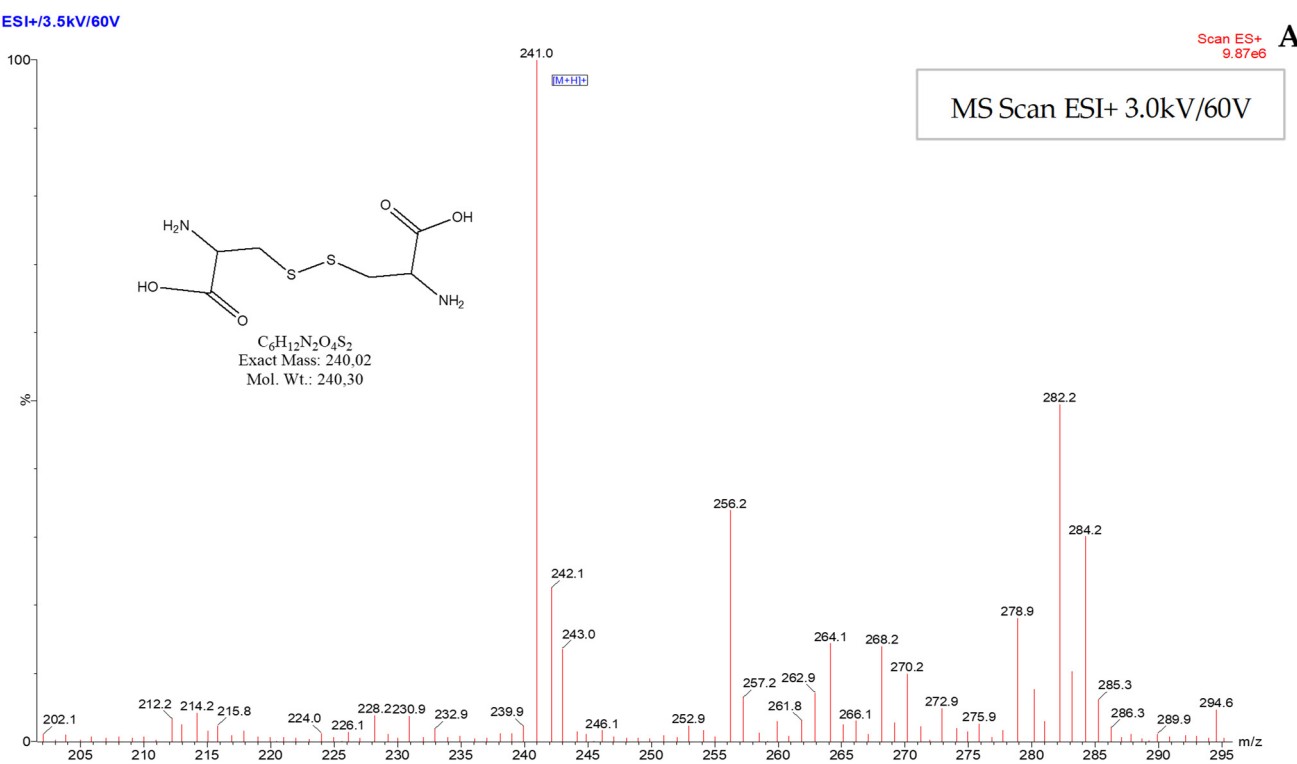

**Figure 7.** *Cont.*

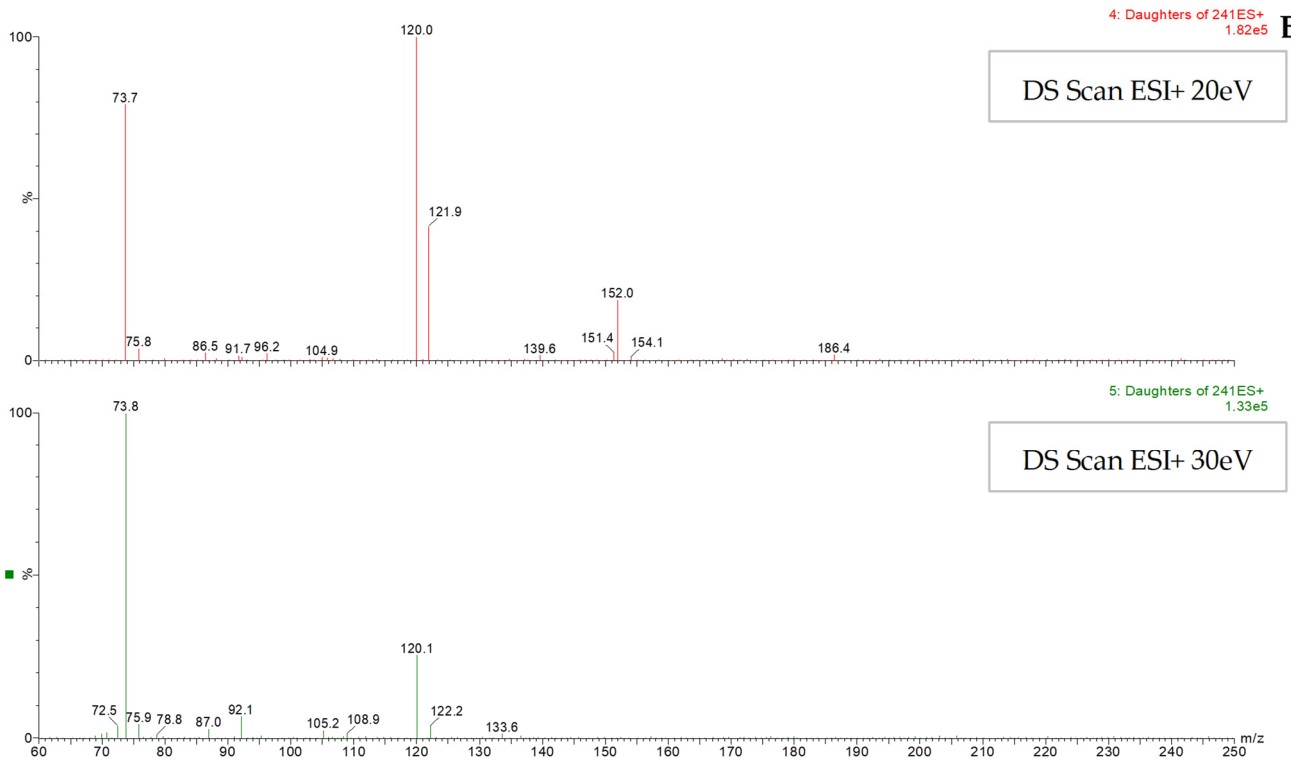

**Figure 7.** Mass spectrum (MS) scans (**A**) and daughter scans (**B**) of the substrate cystine.

Dodecylamine (m/z = 186)

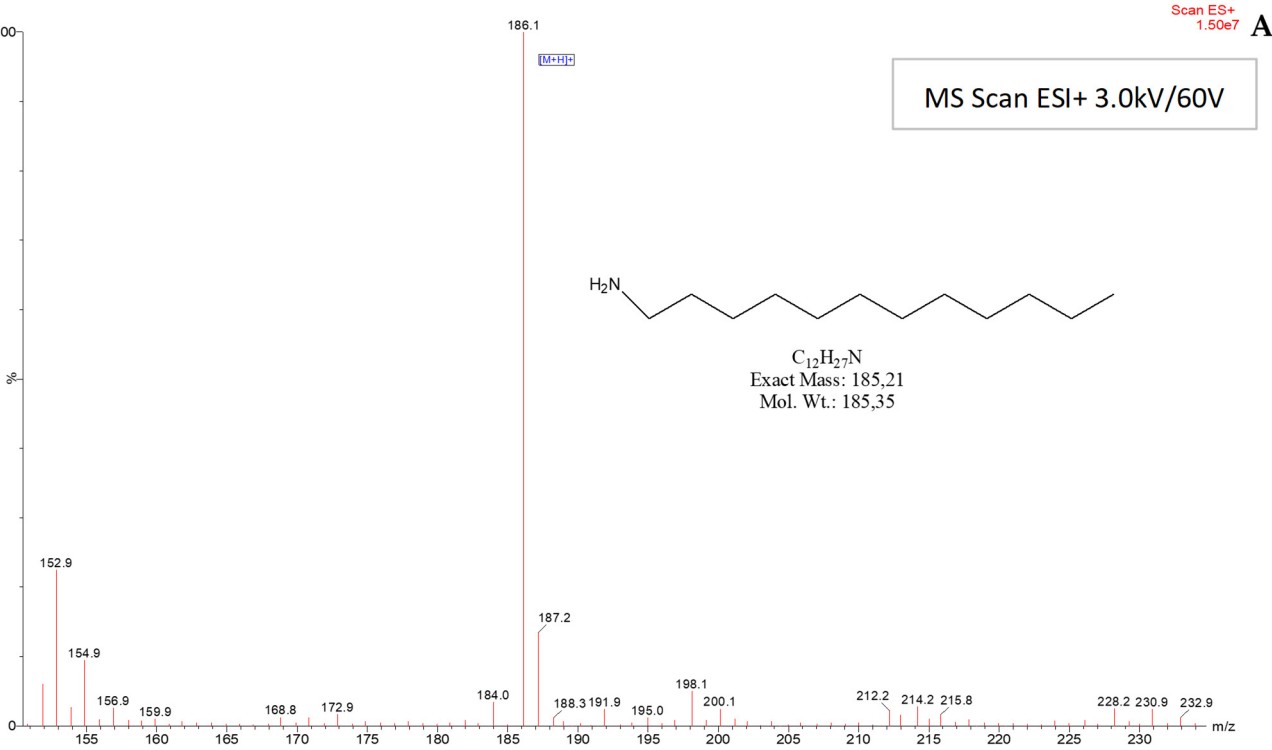

**Figure 8.** *Cont.*

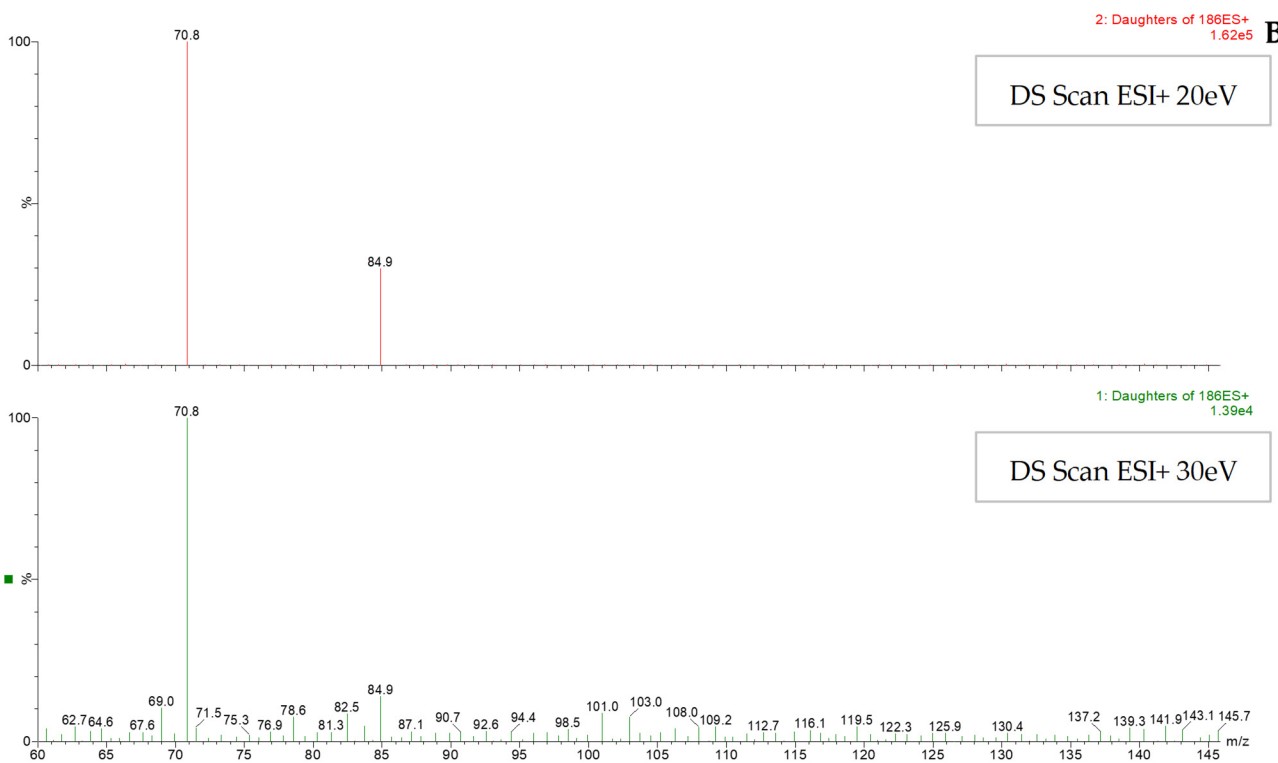

**Figure 8.** Mass spectrum (MS) scans (**A**) and daughter scans (**B**) of the substrate dodecylamine.

### 3.3. Other Chemicals

Tris buffer (Tris-(hydroxymethiyl)aminomethane) was purchased from Merck and sodium chloride (NaCl) from Panreac. Dimethylsulfoxide (DMSO) was acquired from Carlo Erba and 2-propanol, *n*-hexane, and ethanol were from Merck. The eutectic mixture Dowtherm® A was purchased from Sigma-Aldrich as well as methanol, the ninhydrin reagent according to Stahl (17975), and Dragendorff reagent. Tetramethyl orthosilicate (TMOS) was bought from Fluka Analytical and the Protein Assay Dye Reagent concentrate from Bio-Rad.

Three types of buffer solutions were used: (i) Tris buffer 10 mM at pH 6, pH 8, and pH 9; (ii) Tris 10 mM combined buffer with NaCl 150 mM at pH 8 and pH 9; and (iii) acetate buffer 20 mM, pH 6.

All pH measurements were carried out using a Metrohm 744 pH meter.

### 3.4. Docking Studies

Cystine and dodecylamine structures were fitted to the TLL and RML active sites to evaluate their potential as substrates.

All lipases show a common hydrolase fold and a catalytic triad composed of a nucleophilic serine, which is activated via hydrogen bonds as part of a charge relay system, along with the histidine and the aspartate or glutamate residues. In this study, the structures of two lipases, TLL and RML, were used.

There are seventeen crystallographic structures available for *Thermomyces lanuginosus* lipase (TLL) with resolutions between 1.84 and 3 Å. The enzyme structure from the available TLL crystal structure in complex with the lauric acid (PDB code: 4KJX) at 2.7 Å resolution was retrieved from the Protein Data Bank (PDB) databank, although other available crystallographic data showed a better resolution, 4KJX was chosen because in the PDB, the enzyme is complexed with lauric acid, a low molecular-weight molecule very similar with the one in this study. The complex had two chains (A, and B). For this study, we decided to work with only the A chain.

There are four crystallographic structures available for Rhizomucor miehei triacylglyceride lipase (RML) with resolutions between 1.9 and 3 Å. In this work, we used the three dimensional 3TGL structure (3TGL uncomplexed at 1.9 Å resolution). The 3TGL structure is a single polypeptide chain with 269 residues.

The enzyme structures retrieved from the PDB databank were prepared using the protein preparation tools implemented in MOE 2018.10 software.

All crystallographic waters and other non-residue molecules were removed from the coordinate sets, hydrogen atoms were added to this reduced crystal structure, and the protein was protonated to pH 7. The enzyme was then submitted to restrained molecular mechanics refinement using the AMBER99 force field implemented in MOE software. To assess the suitability of these crystal structure-based models, a preliminary validation of the enzyme structures were carried out involving non-covalent re-docking lauric acid to the prepared TLL and RML structures with GOLD software version 5.1.0, with no restrictions on the flexibility of the linking options. For all the docking calculations, standard default settings mode was used: the number of islands was 5, population size of 100, number of operations was 100,000, a niche size of 2, and a selection pressure of 1.1. Gold Score scoring function was used to rank the ligand conformations.

### 3.5. Solubility Assays

The solubility of the substrates L-cystine and dodecylamine in several solvents was tested by weighing a certain mass of the compound, and progressively adding the solvent to be tested until the solution was clear. At each addition of solvent, OD 600 (optic density at 600 nm) was evaluated in order to determine the solubility curves.

The aqueous systems tested were buffer solutions of Tris 10 mM or a combination of Tris 10 mM and NaCl 150 mM, both at pH 8 and 9. The organic solvents tested were dimethylsulfoxide (DMSO), methanol, 2-propanol, *n*-hexane, ethanol, and the eutectic mixture Dowtherm® A.

Due to the low solubility of the substrates in water, a biphasic system of Tris (10 mM) pH 9 and *n*-hexane was also tested.

In these assays, the substrates were dissolved in aqueous and organic solvents to maximum concentrations. Volumes of the two solutions, respectively, cystine and dodecylamine, were combined as the stoichiometry of 1:2 (Cys:Dda). The substrates' final concentrations (mg mL$^{-1}$) were: for Cys 3.6, 2.5, 1.8, 1.3, 0.9, 0.5 and for Dda 4.6, 3.5, 2.3, 1.6, 1.2, 0.7. Tris buffer solution pH 9, was used in a final volume of 1.5 mL. The test tubes were incubated at different temperatures, respectively, 20 °C, 40 °C, 50 °C, and 60 °C.

### 3.6. Immobilization of PPL on Sol-Gel Lenses and Characterization

Sol-gel is a porous material and an optically transparent matrix that enables tailoring of specific necessities in simple methodologies [42]. It has been typically used to encapsulate biomolecules [42,43]. In this type of entrapment, the siloxane polymer chains grow around the enzyme within an inorganic oxide network and the biocatalyst remains accessible to external species due to the pores [42].

In this work, to create the matrix, a sol phase was first produced by adding the alkoxide precursor, tetramethyl-orthosilicate (TMOS) or tetraethyl-orthosilicate (TEOS) with water, a co-solvent and an acid or base catalyst at room temperature [42,43]. Other authors [43] suggest that for lipase immobilization, TMOS allows better activity results. The metal alkoxide ($\equiv$Si–OR) then reacts with the water, initiating the sol-gel reaction [42]. The whole process can be described as the hydrolysis of a silane precursor, followed by cross-linking condensation that causes the development of a SiO$_2$ matrix [43]. The sol-gel events are described in Table 1.

The gelation time of the sol is known [42] to increase with the quantity of organosilane and buffer pH, but decrease with buffer concentration.

In the process of aging, or simply the drying of the material, the network cross-links and the solvent is slowly excluded from the matrix. During this stage, the support suffers

internal alterations like the change of polarity, and viscosity and pore size decrease [42]. If the shrinkage occurs too drastically, the pores collapse and the material fractures, or they became so small that can make the enzyme inaccessible [42,43]. To suppress this problem, additives like trehalose and glycerol can be used to better control the drying process [43].

The methods described and optimized for other enzymes were followed, in order to apply the sol-gel immobilization process to PPL [34].

Briefly, in an Eppendorf was added 96 mg of glycerol (98%), 70 μL of distilled water, 15 μL of HCl (80 mM), and 300 μL of TMOS. The mixture was sonicated for 20 min, at a temperature from 0 to 4 °C. In Table 2, the concise protocols (method A and B) are described. Following the formation of the hydrogel, the shape of the lenses was created using a 96 rounded well microplate for the sol-gel.

In method A (Table 2), the hydrogel was formed directly in the microplate well by independently pipetting 25 μL of sol solution in each one and quickly adding 25 μL of the enzyme solution. Gentle tapping on the microplate was used to unify the mixture.

At this stage, the PPL was added in solutions of three different buffers: Tris (10 mM), at both pH 6 and pH 9, and acetate (20 mM), at pH 6.

In method B (Table 2), the sol was prepared, then 500 μL of PPL solution (in Tris and acetate buffers, pH 6) was added directly in the microtube. Finally, 50 μL of the mixture was pipetted to each microplate well. The lenses were left to dry in the uncovered microplate for a period of no less than 20 h at a controlled temperature of 24 °C.

The quality of the lenses was determined by measuring the diameter, thickness, and weight of the lenses produced by each method.

### 3.7. Sol-Gel Lenses Stability in Biphasic and Co-Solvent Systems

PPL lenses (B1 and B2) produced according to method B (c.f. 3.6) were incubated in a biphasic system of Tris buffer pH 9 and *n*-hexane, and a co-solvent system of DMSO and methanol, in order to determine the resistance of the sol-gel immobilization to the solvents used in these combinations.

Incubations were carried at 20 °C and 60 °C for 24 h. After that period, the diameter, weight, and thickness of the lenses were measured and compared to the values obtained before any incubation.

### 3.8. Biocatalyst Lipozyme® RM Thermostability in Aqueous Media

The thermostability of Lipozyme® RM in different concentrations, 0.5 mg mL$^{-1}$, 2 mg mL$^{-1}$, and 4 mg mL$^{-1}$, was assessed in aqueous media of Tris buffer 10 mM, pH 8, and pH 9.

Three sets of experiments were simultaneously analyzed, the first two monitoring only the evolution of the substrates and the enzyme separately, and the third evaluating the progression the of reaction in aqueous media (Cys 0.23 mg mL$^{-1}$ and Dda 0.26 mg mL$^{-1}$).

The enzyme was tested for three batches of 24 h each at 40 °C, 50 °C, and 60 °C, in a water bath (Julabo SW20) with agitation at 180 rpm.

Samples were collected periodically at 0, 15, 30, 60, 90, 120, 180, 240, 300 min, and 24 h and immediately analyzed through spectrometry at 260 nm. Protein quantification in each sample was carried out using the Bradford method.

After each batch, all media were discharged, the RML particles were rinsed with distilled water, and a new set of assays started. The substrate control was also changed with new solutions at each new reutilization of the enzyme.

### 3.9. Media Design Assays

Liquid reactions were tested using (i) aqueous media of Tris buffer pH 9, (ii) co-solvent systems (Tris:2-propanol; DMSO:MeOH; Tris:DMSO:MeOH, and (iii) biphasic system (Tris:*n*-hexane).

The reactions were conducted at 40 °C in a water bath (Julabo SW20, Seelbach, Germany) with agitation at 180 rpm.

### 3.9.1. Tris Buffer 10 mM, pH 9

After preliminary studies, the effects of pH and temperature were carried out. The thermostability of RML was evaluated in aqueous media, using stock solutions of Cys 0.45 mg mL$^{-1}$ and Dda 0.5 mg mL$^{-1}$ in Tris buffer, in volumes of 2.3 and 2.4, respectively, to a final volume of 4.7 mL.

### 3.9.2. Co-Solvent Systems

The co-solvent systems tested were DMSO:MeOH (1:1), Tris buffer pH 9:DMSO:MeOH (12:1:1), and Tris buffer pH 9:2-propanol (2:1). Stock solutions were prepared for cystine, dissolved in DMSO at either 0.4 mg mL$^{-1}$ or 52 mg mL$^{-1}$, and dodecylamine, in methanol at 0.5 mg mL$^{-1}$ or 62 mg mL$^{-1}$.

For the reactions in the Tris buffer:2-propanol system, one single scheme was tested. In this case, cystine was dissolved at 0.225 mg mL$^{-1}$ in Tris buffer pH 9 and dodecylamine at 0.5 mg mL$^{-1}$ in 2-propanol.

### 3.9.3. Biphasic Systems

In the biphasic system tested, Tris buffer pH 9:*n*-hexane (2:1), cystine was dissolved in Tris buffer pH 9 at 0.225 mg mL$^{-1}$, and dodecylamine was dissolved in *n*-hexane at 0.5 mg mL$^{-1}$. Volumes of each solution were added in different concentrations.

### 3.9.4. Solvent-Free Systems

These solvent-free reactions were carried at 40 °C. At this temperature, one of the substrates, dodecylamine, is liquid, with a melting point of 27–29 °C. One g of dodecylamine was weighed into each reaction tube and all were stabilized at 40 °C to complete melting of the dodecylamine. Then, 300 mg of cystine were added to each tube, along with 500 µL of the eutectic mixture Dowtherm®A. RML and TLL were used in a quantity of 33 mg in the respective reactions. PPL was immobilized using an in house process, afterward, 126 mg of PPL lenses were used.

The tubes were maintained at 40 °C for a period no shorter than 5 h, and up to 24 h. Gently tapping on the tubes was applied periodically to unify the mixture.

### 3.10. Analytic Methods

Thin layer chromatography (TLC) assays conducted on TLC aluminum sheets 20 × 20 cm silica gel 60 F254 (Merck-1.05554.0001) cropped to 2.5 × 10 cm stripes were carried out to follow the bioreaction, substrate consumption, and product formation. The polarity of the eluents chloroform (7): methanol (3) was optimized. The plates were stained either with ninhydrin or Dragendorff reagents. The sample volume applied to the plates was 10 µL.

The measurements of the absorbance at 260 nm were used to monitoring of the evolution of the assays, as this is a general wavelength for peptide species or proteins.

Calibration curves were determined for the substrates individually and in a 1:2 (Cys:Dda) proportion in Tris buffer pH 8 and Tris buffer pH 9.

For product evaluation, controls of the substrates were carried out in each reaction and subtracted from the global absorbance.

Protein quantification was carried out according to the Bradford method [44] adapted to a micromethod [43] for faster multiple sample processing. In this adapted method, 50 µL of dye was added to 100 µL of the sample to be tested in a microplate (Thermo Scientific NuncTM 96 well microplates). The reaction was developed for 2 min and absorbance was read at 595 nm in a microplate reader (Fluostar Omega, BMG LABTACH, Ortenberg, Germany).

Calibration curves of the Bradford reaction with several concentrations of each substrate and of both substrates at 1:2 proportions were carried out.

Interactions between cationic surfactants and eosin have long been reported to cause alterations in the absorption and fluorescence spectra of the dye [45]. In fact, micellization

properties of the surfactants and these dyes can be applied to water treatments through micellar-enhanced ultrafiltration [38].

The interaction of the xanthene dyes has been studied by other authors [46,47], who have demonstrated that eosin and cationic surfactants interact mainly through electrostatic interactions.

Additionally, Chakraborty and co-workers [45] analyzed the spectra response of eosin when exposed to several surfactants, observing a decrease in the absorption maximum of eosin in aqueous solutions, which is usually around 517 nm, and a shift of the absorption maximum to higher wavelengths. When the double tailed cationic surfactant DDA (dido-decyldimethylammonium bromide) was added to eosin, the absorption maximum shifted to 536 nm. For the determination of the eosin maximum absorption wavelength, 300 µL of the eosin solution was added to 500 µL of distilled water. The solution of eosin showed a maximum of absorption around 517 nm, and in the presence of the gemini, the absorption peak moved to 538, which is very similar to the one recorded for DDA [45].

Aqueous solutions of eosin 0.001%, and the gemini lipoaminoacid at several concentrations were prepared.

The colorimetric reaction was produced by adding 500 µL of each to 300 µL of eosin solution. Spectrometry measurements were made using a HITACHI U-2000 Double-Beam Spectrophotometer. A calibration curve was constructed with the solutions of reference gemini, demonstrating a significant correlation between color intensity and product concentration at 538 nm.

### 3.10.1. Equipment

The HPLC-MS/MS sample analysis was conducted in a quadropole, enabling mass selection with the separation of ions of a certain $m/z$ or a scanning mode, with ramped voltages where only a certain $m/z$ crosses the chamber [48].

The first step of tandem mass spectrometry is to choose a $m/z$ from the first spectrometer scanning. Then, this ion is sent to collide and fragment into products that are analyzed by a second spectrometer, producing a "fragmentation scan" or "daughter scan". In this way, the selected $m/z$ from a source spectrum allows a unique fragmentation pattern to be defined. Inverting the method produces a "percursor scan" or "parent scan", where the second spectrometer is set for an ion while the first one scans for masses, detecting the percursor ion [48].

MRM assays are ideal for a very sensitive and specific quantification because it is based on two ion selections, one from each scanning mode, full and daughter scans.

The HPLC analyses were performed on a Waters Alliance 2695 (Waters®, Dublin, Ireland) equipped with a quaternary pump, solvent degasser, auto sampler, and column oven, coupled to a Waters 996 PDA photodiode array detector (Waters®, Ireland).

The tandem mass spectrometer (MS/MS) used was a MicroMass Quattromicro® API (Waters®, Ireland), triple quadrupole type. Compound ionization was performed by an electrospray source in positive mode (ESI+).

Two methods of sample preparation were used: (i) liquid state reactions were evaporated and resuspended in DMSO. For analysis of co-solvent liquid state reactions, the full volume of 14 mL reactions was evaporated in a rotary evaporator, and 2 mL of DMSO was added to the same recipient for total resuspension of the remaining compounds. The same process was used for biphasic liquid state reaction with the exception of the phases having been separated, resulting in the evaporation of smaller total volumes. This analysis was applied only for the reactions carried on with a 27 mg mL$^{-1}$ enzyme concentration; (ii) solid state reaction samples were prepared by collecting 2 mg of the reaction and adding 2 mL of DMSO for a final concentration of 1 mg mL$^{-1}$.

The separation was performed on a normal-phase column (Luna HILIC 1000 × 3.00 mm) at 35 °C using an injection volume of 10 µL. The mobile phase consisted of Milli-Q water containing 0.5% formic acid (A) and acetonitrile (B) at a flow rate of 0.30 mL min$^{-1}$. The eluting

conditions applied were as follows: initial time (0 min) A 5.0%, B 95%; (5 min) A 100.0%, B 0%; (7 min) A 100.0%, B 0%; (10 min) A 5.0%, B 95%; (15 min) A 5.0%, B 95%.

A photodiode array detector was use to scan wavelength absorption from 210 to 600 nm.

MS/MS experiments were performed on Micromass® Quattro Micro triple quadrupole (Waters®, Ireland) with an electrospray in positive ion mode (ESI+) with an ion source at 120 °C, desolvatation temperature of 350 °C, capillary voltage of 3.50 kV, and source voltage of 60 V. The compounds were ionized and spectra of the column eluate were recorded in the full scan mode $m/z$ 60–2000, and SIR $m/z$ 575, 241, 186. Analytical conditions were optimized to maximize the precursor ion signal ([M − H]$^+$). For the MS/MS experiments, different collision energies (eV) were applied to determined characteristic fragments to be used in MRM mode. High purity nitrogen (N$_2$) was used both as a drying gas and as a nebulizing gas. Ultrahigh-purity argon (Ar) was used as the collision gas.

The sample were initially analyzed in full scan mode $m/z$ 60–2000 with ESI+, and SIR $m/z$ 575, 241, 186. Two different collision energies (20 eV and 30 eV) were used to promote fragmentation and determine characteristic fragmentation patterns of the compounds (Table 3).

**Table 3.** Retention times and Multiple reaction monitoring (MRM) transitions determined to each compound.

| Compound | Retention Time/min | [M + H]$^+$ | MRM1 | MRM2 |
|---|---|---|---|---|
| Cystine | 6.70 | 241 | 241 > 74 | 241 > 120 |
| Dodecylamine | 2.39 | 186 | 186 > 71 | 186 > 85 |
| *Gemini* lipoaminoacid | 2.15 | 575 | 575 > 287 | 575 > 286 |

Full scans and daughter scans (20 eV and 30 eV) were carried out for the substrates (Figures 7 and 8) and for the new gemini lipoaminoacid (Figure 9). For these compounds, $m/z$ 241 and 186 ions were selected from the first scans of cystine and dodecylamine, respectively, and both collision energies were also tested with the fragmentation spectrums presented in Figures 7 and 8. For the product, a $m/z$ 575 ion was selected from the first scan (Figure 9) and sent to fragmentation (Figure 9). The fragmentation scan at 30 eV was clearer, pointing to the $m/z$ 287 ion as the most frequent, followed by $m/z$ 240, 345, and 270.

Based on the fragmentation patterns obtained previously, two transitions (parent mass > fragment mass) were determined for each compound and MRM assays run in standard solutions to provide data relating to the retention times and enable the exact quantification of the product in the experimental samples.

To the gemini lipoaminoacid, transitions of 575 > 287 and 575 > 286 were established and in these conditions, the retention time was 2.15 min (Table 3 and Figure 9). Two transitions for the MRM assays were used: MRM1 was the most intense and was used for quantification, while MRM2 served as confirmation. The two transitions contribute with specificity to the MRM since it is not unusual for two different compounds with the same $m/z$ to have one fragment in common, but two fragments in common is rarer. Additionally, the retention point should be the same as well as the MRM1/MRM2 ratio for any concentration. Figure 10 presents the structure of the gemini derived cystine lipoaminoacid.

Gemini lipoaminoacid (m/z = 575)

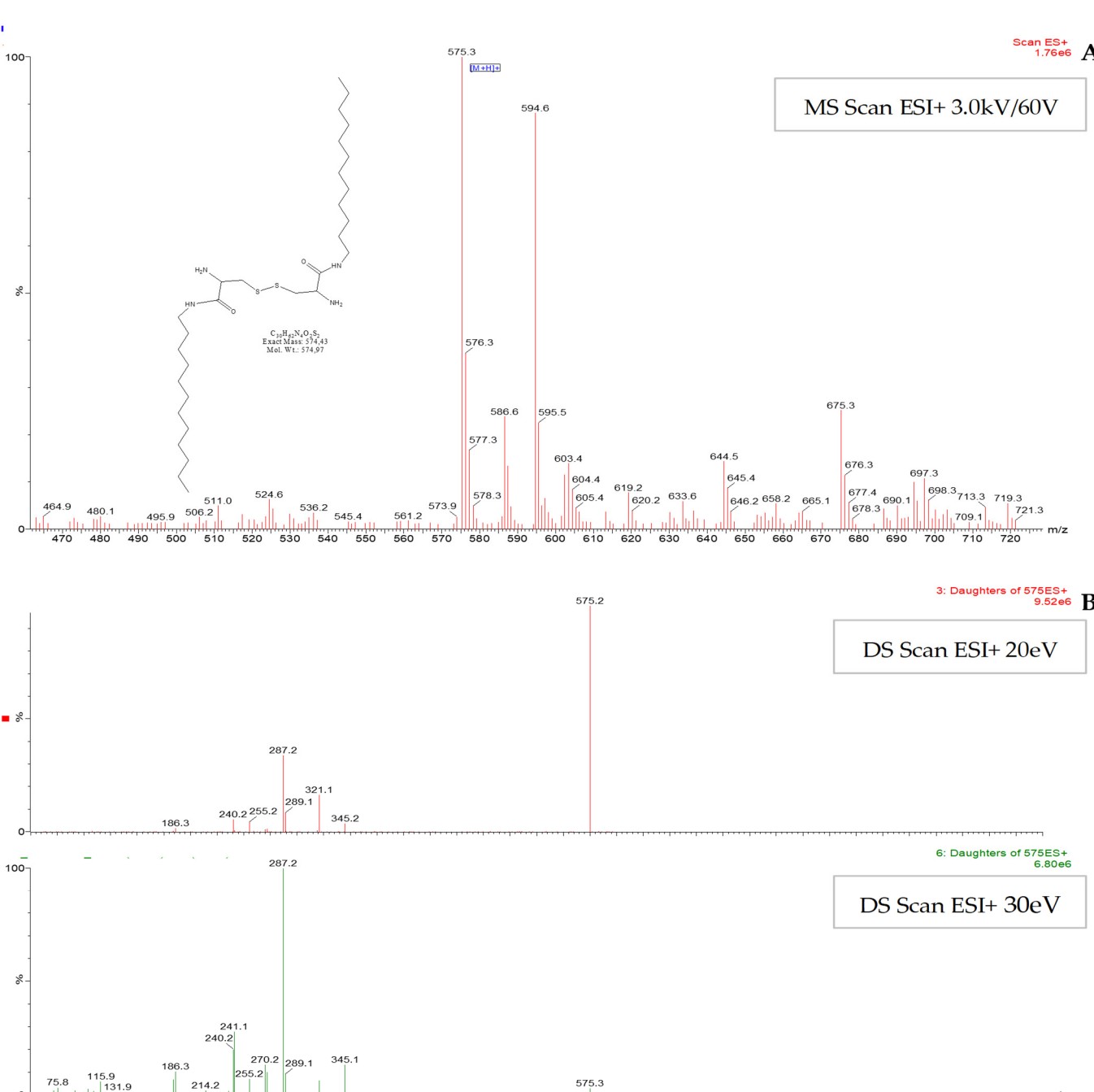

**Figure 9.** Mass spectrum (MS) scans (**A**) and daughter scans (**B**) of the gemini cystine derived lipoaminoacid.

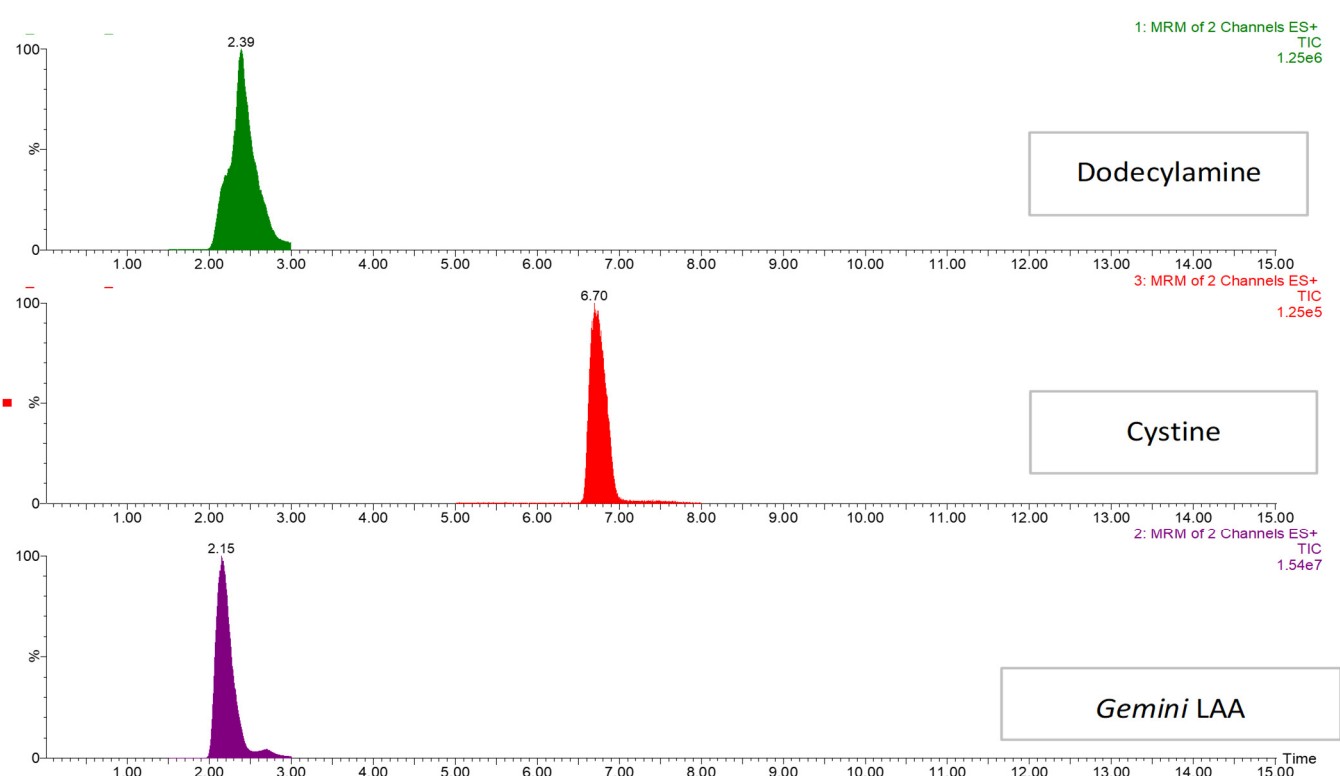

**Figure 10.** Multiple reaction monitoring (MRM) mode chromatograms of 10 mg $L^{-1}$ standard solutions of dodecylamine, cystine, and the gemini lipoaminoacid.

### 3.10.2. Software

Data processing and graphic construction were conducted in either Microsoft Office™ Excel® 2010 or using GraphPad Prism version 5.03 for Windows®, GraphPad Software (San Diego, CA, USA, www.graphpad.com).

For acquisition and processing of HPLC-MS/MS data, MassLynx® version 4.1 was used.

## 4. Conclusions

In conclusion, the docking studies demonstrated the fitting of the substrates cystine and dodecylamine to the active site of the lipases, RML and TLL. In the solvent free media in which PPL was employed, the reaction displayed modifications consistent with the formation of a new surfactant species, visible by texture changes of the reactional volume to the form of foam and identified and quantified by HPLC-MS/MS analysis. In conclusion, this media design was the best to produce LAA among the different bioreaction media studied.

The results presented in this work highlight the potential of the biocatalytic production of lipoaminoacids, a friendly process, as a suitable alternative to current chemical process of multiple extractions, leading to LAA in high purity.

For the first time, using an enzymatic docking and experimental approach, cystine derived gemini cationic biosurfactants were biosynthesized.

**Author Contributions:** All authors have substantially contributed to the conceptualization, methodology, formal analysis, investigation, resources, data curation, writing—original draft preparation, writing—review and editing. All authors have read and agreed to the published version of the manuscript.

**Funding:** The authors are grateful to the FCT—Foundation for Science and Technology, I.P., by the National Funds, under the project UID/DTP/04138/2018, and to FCT for funding the project REDE/1518/REM/2005.

**Institutional Review Board Statement:** Not applicable.

**Informed Consent Statement:** Not applicable.

**Data Availability Statement:** Data available in a publicly accessible repository.

**Acknowledgments:** The authors are grateful to Paulo Madeira for the HPLC-MS/MS analysis, identification, and quantification of the different compounds.

**Conflicts of Interest:** The authors declare no conflict of interest.

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
