# Peer review of "Design of a New Gemini Lipoaminoacid with Immobilized Lipases Based on an Eco-Friendly Biosynthetic Process"

_catalysts, doi:10.3390/catal11020164_

Round 1

Reviewer 1 Report

Unfortunately this manuscript is NOT in the form of a scientific paper to be published in this journal. Instead of the repetitive style used also for elementary concepts authors should format it as usual to be published in this journal thus avoiding to give in additional supplementary material necessary elements for material and methods. Also try to avoid critical assessments of each reference used in the text with long sentences sometimes out of place. Try to avoid to use graphic material in redundant manner (table and graph); a well prepared reaction scheme could be beneficial to the reader. Material should not contain introductive sentences (lines 589-595) or critical assessment. Please give shortly in a preliminar sentece in result also with reference the principle on which the method is based. As authors used 3.9. Media design assays as system to present in results and methods their experimental results this lead to confusion, thermostability experiments reported (in vague terms) in both. Analysis for solvent free reactions is not described. Use a paragraph for Instrumental so to avoid long description of instruments and in specific section report only operative condition. In Analytical section methods of sample preparation are described and only in this section volumes of reactions are reported to describe difference in operativity. Better and more clear system of description of reactions (in media design) should be adopted. I can not find any comments on possible formation of monofunctionalized cystine product. Title seems to be uselessly long. Introduction is giant and unjustified long (lines 38-56, 77-104). I also wonder about usefulness of docking studies in this manuscript, authors should explain briefly how they are useful for the design of production of product. Stence on 203-204 is so vague that is answered also by experimental product formation. Par 2.2 has a redundant preliminar part, no reason for lines 226-229, use and principle for OD600 in solubility assessment is not reported. Solubility assessment is described with redundant style and figure and commented in very long manner without any particular reason while at the end the most important aspect of cooperative effects on solubility are treated in only two lines 270-271. Par 2.2.4 could benefit of some table to avoid long and repetive descriptions. Production expressed in ppb is not familiar and more common mg/ml or mmoles/volume or chemical yields should be used. A conclusion part could be envisaged.

Author Response

Reviewer 1

Instead of the repetitive style used also for elementary concepts authors should format it as usual to be published in this journal thus avoiding to give in additional supplementary material necessary elements for material and methods.

Answer:

Following the suggestion of the reviewer, the content of supplementary elements are now in material and methods, and results, respectively in page 10-11

Also try to avoid critical assessments of each reference used in the text with long sentences sometimes out of place. Try to avoid to use graphic material in redundant manner (table and graph); a well prepared reaction scheme could be beneficial to the reader.

Answer:

The Fig… was rearranjed and now i tis only the graphic. Following the sugestion of reviewer a reaction  sheme (1) was introduced.

Material should not contain introductive sentences (lines 589-595) or critical assessment. Please give shortly in a preliminar sentece in result also with reference the principle on which the method is based.

Answer:

Lines 589-595 were deleted and a new sentence was introduced, as well as Table 1 (this table was in Supplementary Materials and now is in the Experimental section), which sumarizes the sol-gel development materials.

As authors used 3.9. Media design assays as system to present in results and methods their experimental results this lead to confusion, thermostability experiments reported (in vague terms) in both.

Answer:

In “Experimental section 3.9” we consider the methdology developed to study the effect of different solvents in “Media Design Assays (Line 709)”, while in “section 2.2.4. Bioreaction media design”, we present and discuss the obtained results in the bioreaction media, respectively in Aqueous system,  in Co-solvent media, Biphasic systems, Solvent-free media.

Thermostability was evaluated at three different temperatures (40, 50 and 60 ºC) and pH 8.0 and 9.0, in three consecutives batches. In order to clarify this topic the title was changed (page 21 Line 669) as follow “3.8. Biocatalyst Lipozyme® RM thermostability in aqueous media”,.

The results allowed us to conclude that the higher thermostability was attained at 50 ºC and pH 9.0, almost 75 % residual activity remained, at 50 ºC (Page 12, Line 380).

Analysis for solvent free reactions is not described.

Answer:

The solvent free reactions are described in Experimental section (Page 22 and Line 706-710), as follow “At this temperature one of the substrates, dodecylamine, is liquid, with a melting point of 27-29 °C. 1 g of dodecylamine was weighted into each reaction tube and all were stabilized at 40 °C, to complete melting of the dodecylamine” and the obtained results are presented and discussed in Page 15  Line 474- 477.

Use a paragraph for Instrumental so to avoid long description of instruments and in specific section report only operative condition.

Answer:

In  page 23 Line 762-835 was introduced the point 3.10.1 about equipment, as suggested by the reviewer.

I can not find any comments on possible formation of monofunctionalized cystine product.

Answer:

About the possible formation of monofunctionalized cystine product we did not detect this compound. If it was formed it was in such a small quantity that we were not able to detect even in HPLC-MS/MS analysis, that is reason why i tis not mentioned in the paper.

Title seems to be uselessly long.

Answer:

The title was shorten, as follow: from “Design of a new gemini lipoaminoacid based-biosurfactant with immobilized lipases using an eco-friendly biosynthetic process” to “Design of a new gemini lipoaminoacid with immobilized lipases based on an eco-friendly biosynthetic process”

 Introduction is giant and unjustified long (lines 38-56, 77-104).

Answer:

The reviewer 1 state that ntroduction is giant, however reviewer 2 and 3 state that introduction is well prepared and introduces the topic sufficiently and additionnaly reviewer 2 state that it must be complete it. In order to answer to both reviewers, the text in Lines 38-56 and 77-104 was reduced (not complete deleted), as follow:

I also wonder about usefulness of docking studies in this manuscript, authors should explain briefly how they are useful for the design of production of product. Stence on 203-204 is so vague that is answered also by experimental product formation.

Answer:

The goal of the docking studies was the evaluation of  the fitting of the substrates cystine (Cys) and dodecylamine (Dda) to the active site of lipases (RML and TLL). The docking studies demonstrated the fitting of the substrates cystine and dodecylamine to the active site of the lipases, RML and TLL. If these studies proved that there was no fitting, we did not pursue to the experimental work.

Par 2.2 has a redundant preliminar part, no reason for lines 226-229, use and principle for OD600 in solubility assessment is not reported.

Answer:

In page 22, Line 772-724, The following sentence. “The measurements of the absorbance at 260 nm (Eppendorf BioPhotometer) were used to monitoring of the evolution of the assays, as this is a general wavelength for peptide species or proteins.” is expected to explain the use of =D600 in solubility experiments.  

Solubility assessment is described with redundant style and figure and commented in very long manner without any particular reason while at the end the most important aspect of cooperative effects on solubility are treated in only two lines 270-271.

Par 2.2.4 could benefit of some table to avoid long and repetive descriptions.

Answer:

The results of part 2.2.4 (Page 17) are presented in figure 6, if we insert a table, there will be a repetition as reviewer state for Fig. 1 and the respective table.

Production expressed in ppb is not familiar and more common mg/ml or mmoles/volume or chemical yields should be used.

Answer:

As suggested by the reviewer the results are now expressed in mg/mL (Fig. 7) or mg/mL, instead of ppb.

A conclusion part could be envisaged.

Answer:

A conclusion was introduced, in page 28 Line 846-857 as follow: “

In conclusion, the docking studies demonstrated the fitting of the substrates cystine and dodecylamine to the active site of the lipases, RML and TLL. The solvent free media in which PPL was employed, the reaction displayed modifications consistent with the formation of a new surfactant species, visible by texture changes of the reactional volume to the form of foam and identified and quantified by HPLC-MS/MS analysis. In conclusion this media design was the best to produce LAA, among the different bioreaction media studied.

The results presented in this work highlight the potential of the biocatalytic production of lipoaminoacids, a friendly process, as a suitable alternative to current chemical process of multiple extractions, leading to LAA in a high purity.

For the first time using an enzymatic docking and experimental approach cystine derived gemini cationic biosurfactants were biosynthesized.

Reviewer 2 Report

Dear Authors

An interesting study - however, some comments to improve hopefully:

The abstract is very general - please add some more details/results to make it attractive! while deleting some too general parts!

The introduction is well prepared and introduces the topic sufficiently - my only advise would you may add a structural scheme to highlight impoortant compounds and main motives of biosurfactants.

L170 "were observed."

Figure 1 is split over 2 pages; the docking zoom is ok but the A and B structures are too small and the color code shoud be imprvoded and it might be colored according to hydrophilic and hydrophobic residues.

you should explain the cystine abbreviation cys2 as cys is used for cystein and the 2 might be missleading

L316 correct for spaces.

throughout manuscript check chemical annotation and style; "n-hexane" the n need to be italics.

Figure 3; the legend needs to be improved

in Figure 4 I miss a indication of replicates; error bars and detailed legend

L554 italic style of organism designation

software need reference or source descriotion

how was the Tris buffer adjusted to a certain pH the counter ion need to be added. also the acid and base for adjusting.

suppl mat:

- general formatting should be correct and a clear title page is supportive!

- Tab2; the buffers have different pH and or ions; how was the pH adjusted - state; in line B2 the format "-1" need to be corrected.

Author Response

Reviewer 2

The abstract is very general - please add some more details/results to make it attractive! while deleting some too general parts!

Answer:

In the abstract some details/results were introduced in page … Lines…

The introduction is well prepared and introduces the topic sufficiently - my only advise would you may add a structural scheme to highlight impoortant compounds and main motives of biosurfactants.

Answer:

A structural scheme to highlight important compounds, namely lipoaminoacids biosurfactants were introduced in Fig. 1. The main motives of biosurfactants were introduced.

L170 "were observed."

Answer:

We thank the reviewer and it was corrected.

Figure 1 is split over 2 pages; the docking zoom is ok but the A and B structures are too small and the color code shoud be imprvoded and it might be colored according to hydrophilic and hydrophobic residues.

Answer:

The quality of the figures was upgraded as suggested by the reviewer. Figures 1A and 1B have now a better resolution (600dpi's). Hydrophobic regions show as yellow/orange patches ("dry desert" colors, hydrophobic) on the protein surface, hydrophilic regions as green and blue ("wet, watery" colors, hydrophilic).

In the Legend of Figure 1, the following sentence was added “Hydrophobic regions show as yellow/orange patches ("dry desert" colors, hydrophobic) on the protein surface, hydrophilic regions as green and blue ("wet, watery" colors, hydrophilic).”

you should explain the cystine abbreviation cys2 as cys is used for cystein and the 2 might be missleading

Answer:

We thank the reviewer and it was corrected, it is cys and not cys2.

L316 correct for spaces.

Answer:

We thank the reviewer and it was corrected.

throughout manuscript check chemical annotation and style; "n-hexane" the n need to be italics.

Answer:

We thank the reviewer and it was corrected.

Figure 3; the legend needs to be improved

Answer:

We thank the reviewer and it was improved, as follow: “Diameter, weight and thickness parameters of the lens produced according to the method A or B, with the enzyme in Tris buffer (A2 and B1) or acetate buffer (A3 and B2). A1 lens fractured and measurement was not possible. Sol-gel matrices resistance to the organic solvents combinations used as reaction media, by assessing the percentages of diameter and weight, after incubation in buffer Tris pH 9 : n-hexane (C1) and DMSO : MeOH (C2) for 24 hours at 20 and 60 °C. The lenses used in these assays were produced by the methods B1 and B2.”

in Figure 4 I miss a indication of replicates; error bars and detailed legend

Answer:

We thank the reviewer and it was corrected and the standard deviation was indicated, in the legendo of the figure: “Time-course and thermostability of Lipozyme® RML, in Tris buffer 10 mM, pH 8 (I), pH 9 (II). [Lipase RML]  A- 0.5 mg mL-1 ; B – 2 mg mL-1; C – 4 mg mL-1. The SD (Standard Deviation) was ± 0.01, it was so small, that it was impossible to that ist was visible. Each point of the graphic was carried out in triplicates.

L554 italic style of organism designation

Answer:

We thank the reviewer and it was corrected.

software need reference or source description

Answer:

We thank the reviewer and it was introduced.

how was the Tris buffer adjusted to a certain pH the counter ion need to be added. also the acid and base for adjusting.

Answer:

According to the sugestion of reviewer 1 the content of supplementary elements are now in material and methods, and results, respectively in page 10 and page

The pH of the Tris buffer while the acetate buffer was adjusted with acetic acid.

Reviewer 3 Report

The manuscript entitled “Design of a new gemini lipoaminoacid based-biosurfactant with immobilized lipases using an eco-friendly biosynthetic process” reports on the synthesis of lipoaminoacid surfactants using lipases.

It is a very interesting work, that covers the subject on every angle. The authors have done a very thorough planning and research design covering all aspects. The results are clearly presented and support the conclusions. The language needs a quick check as minor errors are present, especially at the Introduction section. What is more important, a paragraph should be added so sum up the conclusions of the presented work. The authors give the conclusions of each part of the study individually but a paragraph to describe the main outcome of the study is definitely missing. It is a different thing to be able to see the outcome of the work in a single sum up paragraph than finding part conclusions after every individual experimental section.

After the minor revisions of adding the conclusions and performing a spell check, I suggest accepting the manuscript for publication.

Author Response

Reviewer 3

The manuscript entitled “Design of a new gemini lipoaminoacid based-biosurfactant with immobilized lipases using an eco-friendly biosynthetic process” reports on the synthesis of lipoaminoacid surfactants using lipases.

It is a very interesting work, that covers the subject on every angle.

The authors have done a very thorough planning and research design covering all aspects.

The results are clearly presented and support the conclusions.

The language needs a quick check as minor errors are present, especially at the Introduction section.

Answer:

We thank the reviewer and it was corrected.

What is more important, a paragraph should be added so sum up the conclusions of the presented work. The authors give the conclusions of each part of the study individually but a paragraph to describe the main outcome of the study is definitely missing. It is a different thing to be able to see the outcome of the work in a single sum up paragraph than finding part conclusions after every individual experimental section.

Answer:

We thank the reviewer and a conclusion was introduced, in page 28 Line 847-857, as follow: In conclusion, the docking studies demonstrated the fitting of the substrates cystine and dodecylamine to the active site of the lipases, RML and TLL. The solvent free media in which PPL was employed, the reaction displayed modifications consistent with the formation of a new surfactant species, visible by texture changes of the reactional volume to the form of foam and identified and quantified by HPLC-MS/MS analysis. In conclusion this media design was the best to produce LAA, among the different bioreaction media studied.

The results presented in this work highlight the potential of the biocatalytic production of lipoaminoacids, a friendly process, as a suitable alternative to current chemical process of multiple extractions, leading to LAA in a high purity.

For the first time using an enzymatic docking and experimental approach cystine derived gemini cationic biosurfactants were biosynthesized.

Round 2

Reviewer 1 Report

Reject